# Superhydrophobic Wood Surfaces: Recent Developments and Future Perspectives

Xianming Gao [†], Mingkun Wang [†] and Zhiwei He *

Anti-Icing Materials (AIM) Laboratory, College of Materials and Environmental Engineering, Hangzhou Dianzi University, Hangzhou 310018, China; 20205109@hdu.edu.cn (X.G.); 221200046@hdu.edu.cn (M.W.)
* Correspondence: zhiwei.he@hdu.edu.cn
† These authors contributed equally to this work.

**Abstract:** Wood is a renewable material that has been widely utilized as indoor and outdoor construction and decoration material in our daily life. Although wood has many advantages (i.e., light weight, high strength, low price and easy machinability), it has some drawbacks that influence dimensional stability, cracking and decay resistance in real practical applications. To mitigate these issues, superhydrophobic surfaces have been introduced to wood substrates, creating superhydrophobic wood surfaces (SHWSs) that can improve stability, water resistance, ultraviolet radiation resistance and flame retardancy. Herein, the recent developments and future perspectives of SHWSs are reviewed. Firstly, the preparation methods of SHWSs are summarized and discussed in terms of immersion, spray-coating, hydrothermal synthesis, dip-coating, deposition, sol-gel process and other methods, respectively. Due to the characteristics of the above preparation methods and the special properties of wood substrates, multiple methods are suggested to be combined to prepare SHWSs rather than each individual method. Secondly, the versatile practical applications of SHWSs are introduced, including anti-fungi/anti-bacteria, oil/water separation, fire-resistance, anti-ultraviolet irradiation, electromagnetic interference shielding, photocatalytic performance, and anti-icing. When discussing these practical applications, the advantages of SHWSs and the reason why SHWSs can be used in such applications are also mentioned. Finally, we provide with perspectives and outlooks for the future developments and applications of SHWSs, expecting to extend the utilization of SHWSs in our daily life and industry.

**Keywords:** wood; superhydrophobic surfaces; preparation methods; versatile applications



## 1. Introduction

Wood is a sustainable and abundant biomaterial that consists of cellulose, hemicellulose, and lignin [1,2]. It possesses a lot of advantages, such as light weight [3], high strength [4], easy machinability [5], renewability [6], thermal insulation [7], aesthetic characteristics [8], low price [9] and environmental friendliness [10]. For thousands of years, wood has been extensively utilized in various applications, including fuels [11], indoor and outdoor construction and decorations [12–15], tools [16], papers [4] and furniture [17]. Due to rich hydroxyl groups, the hydrophilic wood is susceptible to water [3], fire [5], pollutants [18], insects [7], living microorganisms (i.e., bacteria and fungi) [19–21] and weathering factors (i.e., acidic rain and UV lights) (Figure 1) [14,15,22], resulting in its dimensional instability (i.e., swelling and shrinking), cracking, decay and degradation [3,9,11,23]. Thus, wood surfaces are urgently needed to be protected such that they can be stably and safely used in different environmental conditions.

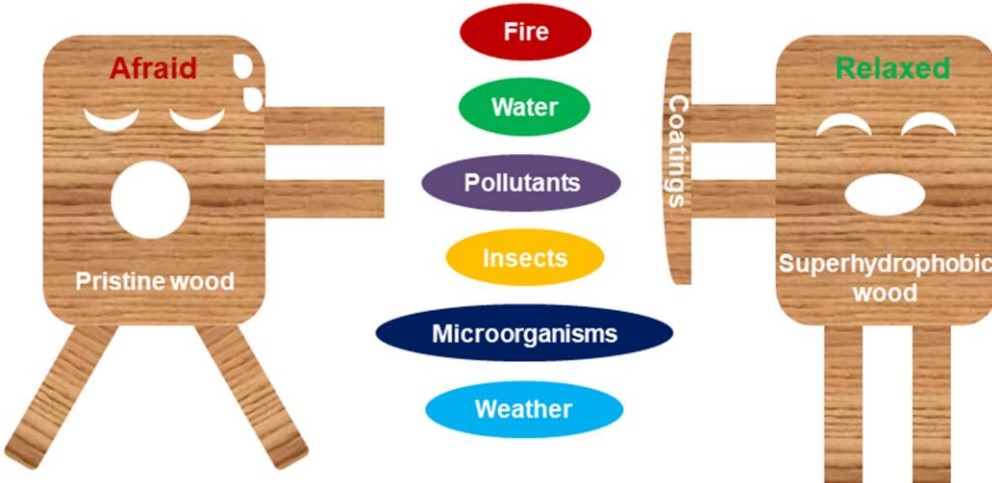

**Figure 1.** Schematic overview of pristine wood and superhydrophobic wood towards environmental factors.

Due to excellent liquid repellency, superhydrophobic surfaces (SHSs) (contact angle > 150° and contact angle hysteresis < 10°) have drawn much interest because of their versatile applications in oil/water separation [24,25], self-cleaning [26], anti-corrosion [27,28], anti-fouling [29] and anti-icing [30,31]. By introducing hierarchical roughness and low surface energy, various substrates can be endowed as SHSs, including wood, metals, ceramics, polymers, glass, fabrics and papers [32–35]. Among them, wood surfaces with hydroxyl groups can be easily decorated with nanomaterials, followed by the chemical modification of low-surface-energy materials. Once superhydrophobic wood surfaces (SHWSs) are achieved, the disadvantages of wood can be mitigated when utilized in real outdoor practical applications.

To the best of our knowledge, there are many reviews on various superhydrophobic surfaces [36–42], and the state-of-the-art review paper on superhydrophobic wood substrates is lacking and urgently needed. Although Li et al. [43], Ramesh et al. [44], and Wei et al. [45] have recently introduced wood-based composites and wood-based superhydrophobic surfaces, a comprehensive review paper particularly focusing on wood substrates by introducing surface superhydrophobicity is still needed. In this review paper, the recent advances in various preparation methods of SHWSs are summarized and discussed, including immersion, spray-coating, hydrothermal synthesis, dip-coating, deposition, sol-gel process and other methods, respectively. Then, the versatile practical applications of SHWSs are introduced in terms of anti-fungi/anti-bacteria, oil/water separation, fire resistance, anti-ultraviolet (UV) irradiation, electromagnetic interference (EMI) shielding, photocatalytic performance and anti-icing, respectively. In the end, the perspectives and outlooks for the developments and practical applications of SHWSs are revealed and proposed. It is believed that the rational design of SHWSs provides new insights into the utilizations of wood materials, and thus extends the real applications of wood materials in our daily life and industry.

## 2. Preparation Methods of SHWSs

Generally, the preparation of SHSs needs to satisfy two necessary requirements: the creation of hierarchical structures and the chemical modification of low-surface-energy materials [46]. To realize surface superhydrophobicity of wood materials, there are two approaches towards preparing SHWSs: (1) the chemical modification of low-surface-energy materials and the simultaneous formation of hierarchical surface roughness (i.e., microstructures and nanostructures); (2) the introduction of hierarchical surface roughness, followed by the chemical modification of low-surface-energy materials [32]. Recently, various preparation methods have been successfully designed and applied on wood mate-

rials with rich hydroxyl groups to achieve SHWSs, including immersion, spray-coating, hydrothermal synthesis, dip-coating, deposition, sol-gel process and other methods. In this section, these methods of preparing SHWSs have been summarized in Table 1, and then they will be separately introduced and summarized in details as well as their own advantages.

**Table 1.** Summary of preparation methods for SHWSs.

| Methods | Types of Wood | Decorated Materials | Modifying Materials | Experimental Conditions | WCA (°) | Ref. |
|---|---|---|---|---|---|---|
| Immersion | Basswood | - | HDTMS, MTMS | Immersed for 6 h, dried at 120 °C for 30 min | $162.9 \pm 2$ | [11] |
| | Poplar wood | Cu | FAS-17 | Pretreated by DA, Immersed at 30 °C for 2 h | 155.7 | [19] |
| | Basswood | Aluminate | - | Immersed at 70 °C, dried for one week at room temperature | 163.6 | [47] |
| | Pine lumber | Cu | FAS | Immersed in NaOH at 70 °C for 10 min | 160 | [48] |
| | Balsa wood | rGO | PDMS | Immersed in GO solution, then reduced to rGO@wood | 152 | [49] |
| | Ash wood | - | APTES | Immersed at 60 °C for 5 h | 161 | [50] |
| | Poplar wood | - | PMS | Immersed at room temperature (RT) for 18 h | 153 | [51] |
| | Wood blocks | $SiO_2$ | PDMS, PMHS | Immersed for 10 min, dried at 100 °C | 164.4 | [52] |
| | Dalbergia sissoo wood | $TiO_2$ | PFOTS | Immersed in a mixture solution for 15 s | $165 \pm 8$ | [53] |
| | Pinus sylvestris wood | $SiO_2$ | OTS | Pretreated with EP, immersed for 10 min | 155.4 | [54] |
| | Balsa wood | $Cu_2O$ | STA | Immersed at RT for 1 h | 153 | [55] |
| | Pinus sylvestris var. mongolica | $Al_2O_3$ | OTS | Pretreated with DA, immersed for 20 h | 152.9 | [56] |
| | Poplar wood | $SiO_2$ | $PCL_{10,000}$ | Immersed for 24 h | $156 \pm 1$ | [57] |
| | Wood substrate | $SiO_2$ | F13-TMS | Immersed for 1.5 h | 156 | [58] |
| | Wood substrate | $SiO_2$, $TiO_2$ | VTES | Immersed at 100 °C for 1.5 h | $153.2 \pm 2.3$ | [59] |
| Spray-coating | Pine wood | $SiO_2$ | Zonyl 8740 | Oven-dried at 105 °C for 12 h | 168.3 | [14] |
| | Birch wood | $SiO_2$ | PDMS | 0.3 MPa at a distance of 10 cm | 170 | [60] |
| | Balsa wood | Soot | PDMS | Sprayed for 3 s | 160 | [61] |
| | European beech wood | AKD wax | | Oven-dried at 45 °C for 10 min | 160 | [62] |
| | Beech wood | CNC@SiO_2@PL rod | PDMS | Dried at 150 °C for 2 h | 157.4 | [63] |
| | Wood substrate | CNC | FOTS | Dried at RT for 15 min | 163 | [64] |
| | Wood substrate | $SiO_2$ | HDTMS | Sprayed for 2 s under pressure of 42 psi | 151.8 | [65] |
| | Wood substrate | $SiO_2$ | N-Boroxine-PDMS | 0.24 MPa at a distance of 12 cm | 160.9 | [66] |
| | Wood substrate | PDVB NTs, $SiO_2$ | PFS | 0.3–0.4 MPa at a distance of 10–15 cm | 157.7 | [67] |
| | European beech wood | AKD wax | | Using a 0.15 mm nozzle | 166 | [68] |
| | Masson's pine, pecan wood | $Cu_2O$ | STA | Sprayed at a 30° angle and a distance of 60 cm | 155 | [69] |
| | Wood blocks | $TiO_2$ | PMC | Sprayed for 10 s, cured at 103 °C for 8 h | $152.2 \pm 2.6$ | [70] |
| | Balsa wood | $SiO_2$ | POTS | Sprayed for 6 s | 151.2 | [71] |
| | Wood substrate | $CNC/SiO_2$ | PFTS | Pretreated with adhesive, dried at RT | >159 | [72] |
| | Chinese fir wood | $PS/SiO_2$ | PDMS | Sprayed for 10 s, baked at 100 °C for 2 h | 150 | [73] |
| | Styrax tonkinensis wood | ZnO | STA | 0.2 MPa at a distance of 10 cm | 154.1 | [74] |
| | Wood substrate | $SiO_2$ | DTMS | Sprayed at a distance of 20–25 cm | $162.3 \pm 1.1$ | [75] |
| | Wood substrate | NCF | PFOCTS | 0.2–0.4 MPa at a distance of 45–60 cm | 161 | [76] |
| | Pine wood | CNC | FOTS | 0.2–0.4 MPa at a distance of 30–60 cm | 162 | [77] |
| Hydrothermal synthesis | Polar wood | $WO_3$ | SDS | 120 °C for 6 h | 152 | [1] |
| | Wood blocks | ZnO | STA | 90 °C for 6 h | 154 | [5] |
| | Spruce wood | $TiO_2$ | SDS | 75 or 90 °C for 1 h | >150 | [7] |
| | Eucalyptus wood | $TiO_2$ | VTES | 100 °C for 6 h | 153 | [20] |
| | Poplar wood | $Ag-TiO_2$ | FAS-17 | 90 °C for 5 h | 153.2 | [78] |
| | Wood slices | $MnFe_2O_4$ | FAS-17 | 120 °C for 8 h | $156 \pm 1$ | [79] |
| | Sapwood | ZnO | Palmitoyl chloride | 90 °C for 3 h | 155 | [80] |
| | Wood substrate | ZnO | DTMS | 90 °C for 3 h | 156 | [81] |
| | Polar wood | $TiO_2$ | SDS | Reheated at 70 °C for 4 h | 154 | [82] |
| | Poplar wood | $TiO_2$ | OTS | 90 °C for 5 h | 152 | [83] |
| | Poplar wood | $TiO_2$ | Fluoroalkyl silane | 90 °C for 5 h | 152.9 | [84] |
| | Birch veneer | $WO_3$ | OTS | 90 °C for 12 h | 152 | [85] |
| | Birch veneer | $WO_3$ | OTS | 110 °C for 24 h | 150.1 | [86] |
| | Pinus wood | $Cu_2(OH)_3Cl$ NPs | STA | 70 °C for 10 min | $151 \pm 3$ | [87] |
| | Poplar lumbers | $\alpha$-FeOOH | OTS | 80 °C for 0.5, 1, 2, 4 h | 158 | [88] |
| | Poplar sapwood | $Cu_2O$ | FAS-17 | 180 °C for 2 h | 153.8 | [89] |
| | Poplar wood | Ti/Si | VTES | 100 °C for 6 h | 152.7 | [90] |
| Dip-coating | Spruce veneers | $SiO_2$ | POTS | Dip-coated in sol, stirred for another 5 h | 151.8 | [3] |
| | Juvenile teakwood | AKD | - | Heating to form $\beta$-ketoester bond | $150 \pm 2$ | [9] |
| | Chinese Cunninghamia lanceolata | $SiO_2$ | PMHS | Dipped for 5 min, air-dried for 1 min | $154.1 \pm 2.1$ | [12] |
| | Cathay poplar wood | $CoFe_2O_4$ | FAS | Dipped for 3 min, air-dried for 20 min | 158 | [22] |
| | Chinese fir wood | $SiO_2$ | PDMS | Dipped for 10 min, dried at 103 °C for 1 h | 152 | [91] |
| | Wood substrate | $Al_2O_3$ | PDMS | Dipped for 30 s, dried at 50 °C for 20 min | 154 | [92] |
| | Poplar wood | $CeO_2$ | OTS | Dipped for 20 min, dried at 80 °C for 1 h | 152 | [93] |
| | Chinese white pine | $SiO_2$ | PDMS, FAS | Dipped for 5 min, dried at 103 °C for 15 min | >150 | [94] |
| | Poplar sapwood | Si-sol | PDMS | Evacuated to −0.09 MPa, dried at 103 °C for 3 h | 151.6 | [95] |
| Deposition | Poplar wood | - | PDMS | CVD, hydrolysis reaction | 157.3 | [96] |
| | Balsa wood | - | MTMS | Vapor deposition | 151.8 | [97] |
| | Birch veneers | Silicone nanofilaments | TCOS | CVD for 3 h | 156 | [98] |
| | Golden chinkapin wood | - | PFE | Plasma deposition at a working pressure of 1.0 Torr | $161.2 \pm 1.5$ | [99] |
| | Wood substrate | Candle soot | Paraffin wax | Evaporation and deposition | 162 | [100] |
| | Chinese Cunninghamia lanceolata | $SiO_2$ | PMHS | Deposition of 75 nm $SiO_2$ NPs | 158.2 | [101] |
| | Larch species wood | ZnO | Octadecanoic acid | Maintained at 70 °C for 5 h | 156 | [102] |
| | Radiata pine | Cu | Perfluorocarboxylic acid | Magnetron sputtering | 154 | [103] |

**Table 1.** *Cont.*

| Methods | Types of Wood | Decorated Materials | Modifying Materials | Experimental Conditions | WCA (°) | Ref. |
|---|---|---|---|---|---|---|
| Sol-gel | Walnut wood | - | PDMS | Immersed for 12 h | 162.4 | [104] |
| | Chinese fir wood | Silica | HDTMS | Dipped for 30 min to deposit silica | 152 | [105] |
| | Poplar lumbers | SiO$_2$ | POTS | Maintained at RT for 6 h | 164 | [106] |
| | Poplar sapwood | TiO$_2$ | PDMS | In-situ deposition, EB radiation curing | 165.7 | [107] |
| | Pine blocks | Silica | Aerosil R-972 | Placed for 10 min, dried in air for 24 h | 152 | [108] |
| | Chinese fir wood | Silica | HDTMS | Immersed for 30 min, dried at 80 °C for 10 h | 150 | [109] |
| | Eucalyptus wood | SiO$_2$ | PFDS | Stirred at 50 °C for 4 h, dried at 70 °C for 10 h | 159 | [110] |
| Assembly | Norway spruce | ZnO | Carnauba wax | Layer-by-layer | 155 | [15] |
| | Southern pine sapwood | ZnO | STA | Self-assembly | >150 | [111] |
| | Poplar wood | TA–Fe$^{III}$ | Octadecanethiol | Multistep assembly | 156 | [112] |
| | Populus ussuriensis Kom. | SiO$_2$ | POTS | Layer-by-layer | 161 | [113] |
| | Poplar wood | TiO$_2$ | POTS | Layer-by-layer | 161 | [114] |
| Brushing | European beech | - | GSE | Dried for 5 min, repeated 5 times | 159 ± 2 | [16] |
| | Pinewood | ZnO, TiO$_2$ | Acetic acid | Brushed for 3 layers, dried at 100 °C for 24 h | >150 | [115] |
| | Chinese fir wood | SiO$_2$, TiO$_2$ | PTES | Curd at RT | 152 | [116] |
| Drop-coating | Poplar lumbers | PS/silica | OTS | 1 mL cm$^2$, dried at RT | 153 ± 1 | [117] |
| | Wooden sheet | TiO$_2$ | H-PDMS | Kept at RT for 30 min, cured at 120 °C for 10 h | 155.5 ± 1 | [118] |
| | Poplar lumbers | SiO$_2$ | OTS | Oven-dried at RT for 12 h | 159 | [117] |
| Grafting | Chinese fir wood | - | PFOEMA | ATRP | 156 | [119] |
| | Radiata pine | - | Stearoyl chloride | The ester linkage | 152 | [120] |
| Casting | Chinese fir wood | Polymer latex | - | Drop-casting, dried at 30 °C | 152 | [121] |
| | Chinese fir wood | Nano fumed silica | Acetic acid | Heating at 120 °C for 2 h | 160 | [122] |
| Impregnation | Pine wood | Cu(OH)$_2$ | Dodecanethiol | Dried in an oven | 154 | [123] |
| | Poplar specimens | TiO$_2$ | Maleic rosin | Dried in an oven at 180 °C for 4 h | 157 | [124] |
| | Poplar sapwood | Silica | Silicone oil | At a vacuum of 0.01 MPa for 1 h, a subsequent pressure of 0.5 MPa for 1 h | 154.8 | [125] |
| | Pinewood | Copolymers | - | Impregnated for 2 h, heated at 140 °C for 2 h | >150 | [126] |
| | Masson's pine wood | Cu$_2$(OH)$_3$Cl | PF and STA | Impregnated at RT for 4 h, immersed at RT for 2 h | 163 | [127] |
| Solvent-based methods | Balsa wood | PVDB | - | A solvothermal method, heated at 100 °C for 24 h | 160 | [128] |
| | Chinese fir wood | SiO$_2$ | VTES | Alkali-driven method | 156.6 | [129] |
| Template methods | Populus ussuriensis Kom. | Fe$_3$O$_4$ | PDMS | Soft lithography | 152 ± 2 | [130] |
| | Populus ussuriensis Kom. | PVB/SiO$_2$ | OTS | Nanoimprint lithography | 160 | [131] |
| | Ash wood | PVB/SiO$_2$ | OTS | Replication | 155 | [132] |

## 2.1. Immersion

Immersion is a facile and cheap method, which does not need complex operations or special devices. Because of surface hydrophilicity, wood is easy to react with chemicals in solutions [133,134]. Usually, there are two approaches towards preparing SHWSs with the immersion methods: (a) One-step immersion method [135,136]. That is, wood substrates are directly immersed either in the solutions of low-surface-energy materials or in mixture solutions of nanoparticles and low-surface-energy materials. (b) A two-step method [137]. Wood substrates are firstly immersed in the suspension of nanoparticles. After being immersed in above suspension solutions, SHWSs can be easily obtained with further chemical modification of low-surface-energy materials [19,47–49,104].

As for one-step immersion, wood can be immersed in solutions of low-surface-energy materials or mixture solutions of nanoparticles and low-surface-energy materials [50,138]. As we all know, both hierarchical structures and low surface energy on wood substrates are necessary for the preparation of SHWSs. The reason why SHWSs can be obtained by immersing wood substrates in the solutions of low-surface-energy materials is that micro/nanostructures can be simultaneously formed during an immersion process [32]. For example, Ou et al. achieved SHWSs by immersing wood substrates (i.e., balsawood, pine wood and basswood) in a composite silane solution consisting of hexadecyltrimethoxysilane (HDTMS) and methyltrimethoxysilane (MTMS) [11]. Liu et al. fabricated SHWSs by using potassium methyl siliconate via a convenient solution-immersion method [51]. Lin et al. realized the production of SHWSs by immersing wood in a modifier solution of n-hexane, PMHS and Kastredt catalyst [10]. Furthermore, when aqueous mixture solutions of nanoparticles and low-surface-energy materials are utilized, SHWSs can also be achieved by one-step immersion. Jia et al. placed wood substrates in a homogeneous reactant solution (i.e., perfluorooctyltriethoxysilane (PFOTS) and BiOCl) for 1.5 h and achieved SHWSs (Figure 2a) [139]. Yue et al. obtained RTVSR/SiO$_2$ coated superhydrophobic wood by immersing wood into the modifier solutions of poly(methylhydrogens)siloxane (PMHS),

vinyl-terminated polydimethylsiloxane (Vi-PDMS) and SiO$_2$ nanoparticles for 10 min and drying wood at 100 °C for 60 min [52]. Thus, the one-step immersion method as a facile approach is suitable for preparing SHWSs.

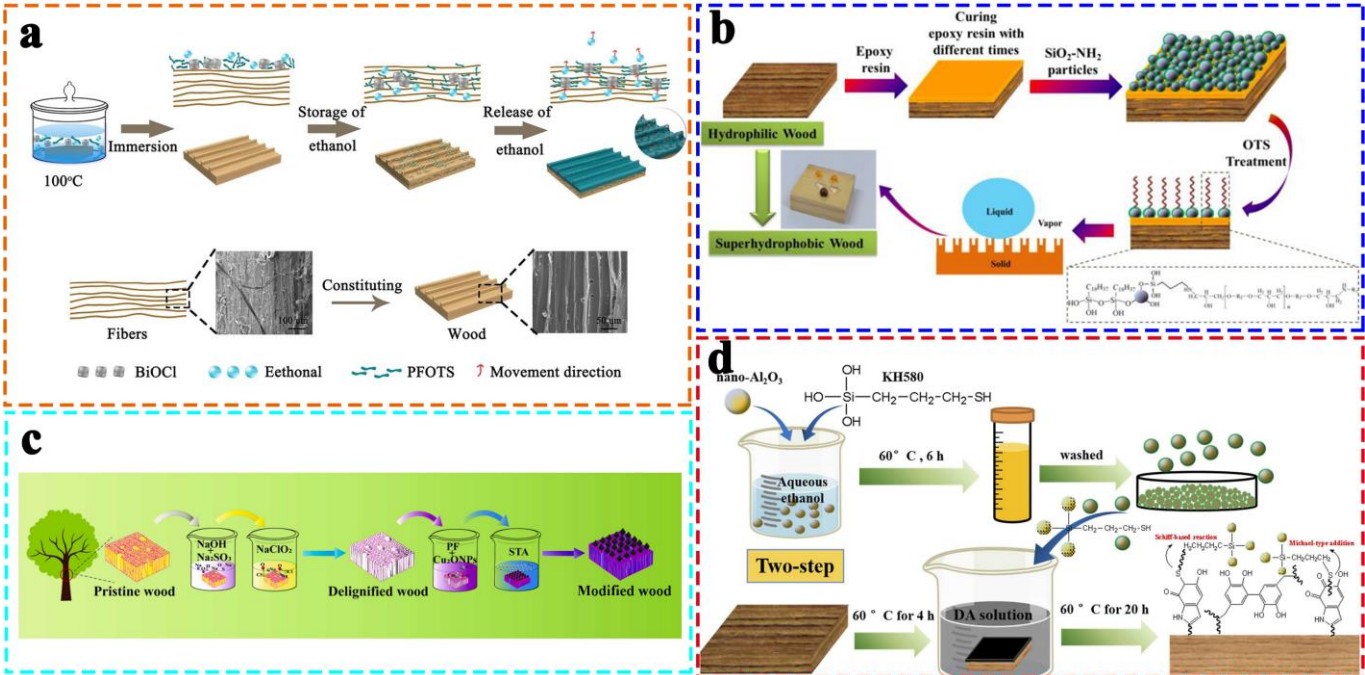

**Figure 2.** Schematic overview of preparing SHWSs by immersion: (**a**) Fabrication of self-sealing SHWSs by introducing hierarchical BiOCl on wood; (**b**) fabrication of SHWSs with amino-functionalized nano-silica particles, epoxy resin and octadecyltrichlorosilane (OTS); (**c**) fabrication of SHWSs with Cu$_2$O nanostructures, stearic acid and phenol formaldehyde resin; (**d**) fabrication of SHWSs with polydopamine, 3-mercaptopropyltriethoxysilane, nano-Al$_2$O$_3$ and OTS. Panel (**a**) reproduced with permission from ref. [139], copyright 2019, Elsevier. Panel (**b**) reproduced with permission from ref. [54], copyright 2021, NC State University. Panel (**c**) reproduced with permission from ref. [55], copyright 2022, Elsevier. Panel (**d**) reproduced with permission from ref. [56], copyright 2021, Elsevier.

As for the two-step method, the hierarchical roughness is usually achieved first, followed by the chemical modification of low-surface-energy materials [53]. For instance, Kang et al. obtained SHWSs by separately immersing wood substrate in the epoxy resin solution and the SiO$_2$-NH$_2$ particle suspension followed by treatment with 2 wt.% octadecyltrichlorosilane solution (Figure 2b) [54]. Xia et al. prepared SHWSs by using Cu$_2$O nanostructures and phenol formaldehyde resin via an immersion process, followed by the chemical modification of 4.8 wt.% stearic acid solution (Figure 2c) [55]. Yang et al. used nano-Al$_2$O$_3$, KH580 and polydopamine (PDA) to obtain hierarchical roughness on wood substrates, and then modified the resulting wood substrates with 2 wt.% octadecyltrichlorosilane (OTS) solution to realize low surface energy (Figure 2d) [56]. Gao et al. coated silica particles on wood by using an immersion method and then fabricated poly($\varepsilon$-caprolactone) film on the resulting wood surface by using a simple pipetting process [57]. Compared with the one-step immersion method, the surface roughness can be enhanced by using the two-step immersion method, thereby improving the surface superhydrophobicity of wood substrates.

To accomplish the immersion process, wood substrates should be immersed in solutions. Compared with other substrates (i.e., metal, glass and ceramics), however, most wood materials are lighter than water, which need to be pressed in the immersion solution [58,59]. In addition, the immersion method can be scaled up for the preparation of SHWSs. Therefore, the immersion method as a low-cost approach is suitable for the fabrication of SHSs (i.e., SHWSs).

### 2.2. Spray-Coating

Spray-coating is a simple, low-cost, substrate-independent, scalable and efficient method to prepare very thin coatings [4,140–145]. When the spray-coating method is used for the preparation of SHWSs, it is supposed to simultaneously form hierarchical roughness and low-energy surfaces. After a curing and/or drying process (i.e., the evaporation of solvents), the dispersion of the desired solutions can be applied on the wood substrate to form superhydrophobic coatings [14,60–63,146–148]. In addition, the thickness of superhydrophobic coatings can be easily adjusted by many factors, including the concentration of dispersion [64], the pressure of spray gun [149], the spraying time [65], the repeated times [66], the distance between wood substrate and spray gun [67] and the drying temperature [68].

For example, Zhan et al. sprayed a solution of $Cu_2O$ nanoparticles/phenol formaldehyde (PF) by using a spray gun on the Masson's pine and pecan wood samples with a distance of 60 cm, followed by the treatment with 5% solution of stearic acid in ethanol at ambient temperature for 2 h [69]. Tu et al. spray-coated a dispersion of $TiO_2$/perfluoroalkyl methacrylic copolymer (PMC)/water (i.e., the mass ratio is 1:1:25) on wood substrate for 10 s, and air-dried for 10 min, which was repeated for 3 times to realize a superhydrophobic coating with a desired thickness [70]. Huang et al. prepared a paint mixture solution consisting of lignin-coated cellulose nanocrystals (L-CNC) particles and polyvinyl alcohol (PVA) particles, and then successfully realized superhydrophobic coatings on wood substrate by spraying a L-CNC/PVA composite paint mixture followed by modifying via chemical vapor deposition (Figure 3a) [150]. Che et al. prepared a dispersion of hydrophobic 1H,1H,2H,2H-perfluorooctyltrichlorosilane/$SiO_2$ nanoparticles (POTS-$SiO_2$), and then evenly spray-coated this dispersion on a flexible Janus wood membrane to achieve SHWSs (Figure 3b) [71].

Although the spray-coating method is a substrate-independent approach towards preparing SHWSs, the adhesion between superhydrophobic coatings and wood substrates should be considered such that the durability of SHWSs can be enhanced. To increase the adhesion between coatings and wood substrates, an adhesive layer is usually introduced on wood substrates before a spray-coating process [72,73]. For instance, Huang et al. treated wood substrate with an adhesive layer of commercial quick-drying transparent topcoat, and then achieved SHWSs by spraying a hydrophobic CNC-ethanol suspension after drying for 15 min at room temperature (Figure 3c) [64]. Tuong et al. treated wood substrates with a 50 wt.% Epoxy#3021 solution of two components (Part A and Part B), and then spray-coated a suspension of hydrophobic ZnO nanoparticles on the epoxy-pretreated Styrax tonkinensis wood by using a spray gun (i.e., 0.2 MPa compressed air) at a distance of ~10 cm (Figure 3d) [74]. Yang et al. firstly spray-coated an epoxy resin/acetone solution onto 1-Tetradecanol (TD)-decorated delignified wood (DW) substrate, and then further deposited $SiO_2$ to obtain the superhydrophobic TD/DW composite (Figure 3e) [147]. Xue et al. spray-coated a $SiO_2$@HDTMS superhydrophobic spray solution of $SiO_2$, an inorganic aluminum phosphate (AP) adhesive layer and Hexadecyltrimethoxysilane (HDTMS) on various substrates (i.e., wood) by using a spray gun with an orifice diameter of 0.8 mm under an exhaust pressure of 42 psi (Figure 3f) [65].

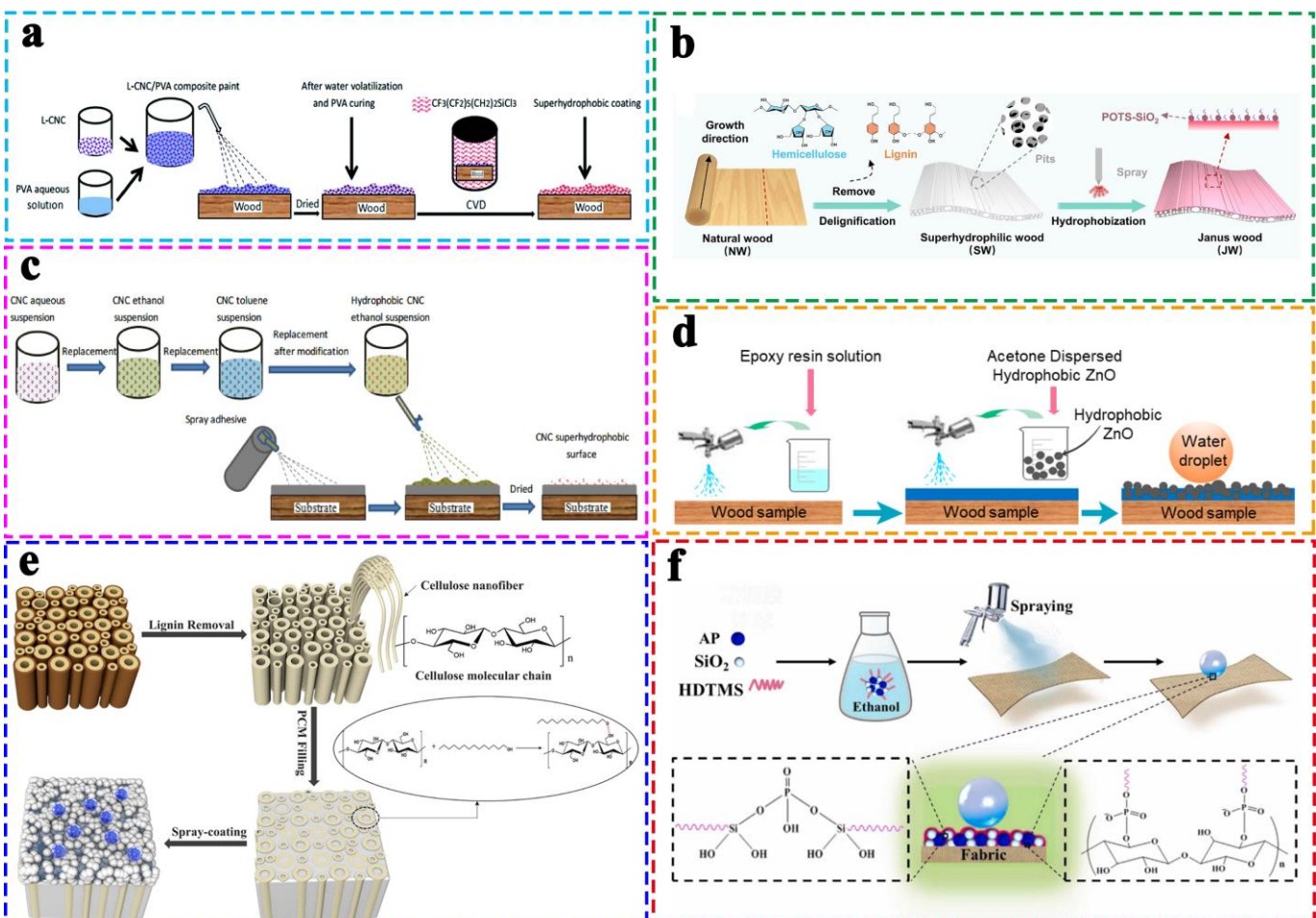

**Figure 3.** Schematic overview of preparing SHWSs by spray-coating: (**a**) Preparation of L-CNC/PVA composite superhydrophobic coating on wood substrate, (**b**) preparation of superhydrophobic flexible Janus wood membrane, (**c**) fabrication of CNC superhydrophobic coating, (**d**) preparation of superhydrophobic ZnO coating on Styrax tonkinensis wood, (**e**) fabrication of superhydrophobic TD/DW composite PCMs, (**f**) preparation of superhydrophobic SiO$_2$@HDTMS coatings. Panel (**a**) reproduced with permission from ref. [150], copyright 2017, the Royal Society of Chemistry. Panel (**b**) reproduced with permission from ref. [71], copyright 2023, Elsevier. Panel (**c**) reproduced with permission from ref. [64], copyright 2018, Elsevier. Panel (**d**) reproduced with permission from ref. [74], copyright 2019, the authors. Panel (**e**) reproduced with permission from ref. [147], copyright 2020, Elsevier. Panel (**f**) reproduced with permission from ref. [65], copyright 2022, Elsevier.

Of course, the spray-coating method can be utilized by combining with other preparation methods to fabricate SHWSs, such as UV polymerization [75], chemical vapor deposition (CVD) [76,77,151], plasma [152] and immersion [153]. Spray-coating, as a widely used approach, is not limited by surface structures or the shapes of substrates, which is suitable for various substrates (i.e., wood, glass, metals, ceramics and papers). Thus, the spray-coating method can be efficiently used for the preparation of SHWSs.

### 2.3. Hydrothermal Synthesis

Hydrothermal synthesis is a method that nanoparticles or surface crystal morphologies can be realized in aqueous solutions at high vapor pressure [140]. During a hydrothermal process, the reaction temperatures (i.e., 85 °C [154,155], 90 °C [1,5,78,156], 100 °C [20] and 120 °C [1,79]) and the reaction time (i.e., 2 h [154], 5 h [78], 6 h [1,5,20,156], 8 h [79] and 12 h [155]) have been chosen for the reaction, which greatly influence the formation of crystalline phases around the melting point of reactants. This approach can be em-

ployed to create hierarchical micro/nanostructured crystals on the desired wood substrates when preparing SHWSs, including $MnFe_2O_4$ [79], $CoFe_2O_4$ [156], ZnO [5,80,81,155,157], $TiO_2$ [7,20,78,82–84], $WO_3$ [1,85,86], $Cu_2(OH)_3Cl$ [87], $\alpha$-FeOOH [88], $Cu_2O$ [89] and Ti/Si composite [90].

Wang et al. successfully planted lamellar $MnFe_2O_4$ on wood substrate through the association of hydrogen bonds via one-pot hydrothermal method (Figure 4(a1)), and obtained SHWSs by the chemical modification of fluoroalkylsilane (Figure 4(a2,a3)) [79]. Gan et al. prepared a hydrophilic layer of $CoFe_2O_4$ nanoparticle on the wood substrate via a hydrothermal method, and realized a superhydrophobic coating on wood substrate by the chemical modification of octadecyltrichlorosilane (OTS) (Figure 4b) [156]. Gao et al. fabricated a durable $Cu_2O$ microsphere superamphiphobic film on wood substrate via a hydrothermal method, followed by the chemical modification of (heptadecafluoro-1,1,2,2-tetradecyl)trimethoxysilane (FAS-17) [89]. Sun et al. achieved the in situ synthesis of $WO_3$ nanoparticles on a wood substrate via a two-step hydrothermal process, and successfully obtained a photochromic and superhydrophobic wood substrate [1].

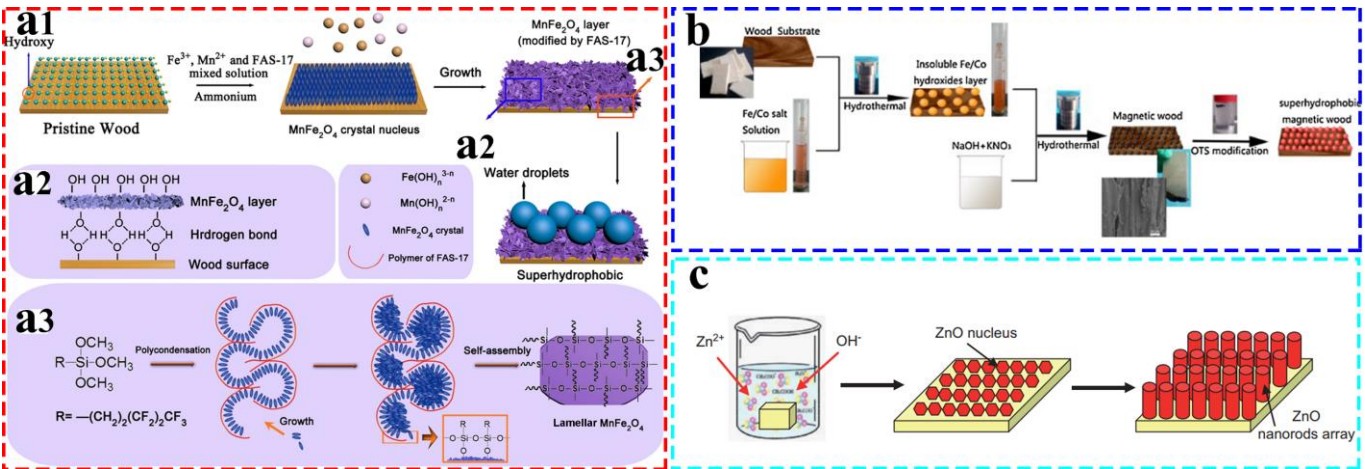

**Figure 4.** (**a1**–**a3**) Possible schematic illustration of the preparation of the $MnFe_2O_4$/wood composite. (**a2**,**a3**) the combination mechanism for the $MnFe_2O_4$ layer and the wood surface and the formation mechanism of the hydrophobic lamellar $MnFe_2O_4$. (**b**) Preparation process of superhydrophobic and magnetic wood. (**c**) Scheme of fabricating superhydrophobic coating of ZnO nanorod array on wood substrate. Panel (**a**) reproduced with permission from ref. [79], copyright 2016, the authors. Panel (**b**) reproduced with permission from ref. [156], copyright 2015, Elsevier. Panel (**c**) reproduced with permission from ref. [154], copyright 2012, Walter de Gruyter Berlin Boston.

Among the above nanomaterials, the preparation of ZnO and $TiO_2$ are widely utilized by the hydrothermal method. For example, Fu et al. investigated the formation of a superhydrophobic ZnO nanorod array on wood substrate via a cosolvent hydrothermal method at 85 °C for about 2 h (Figure 4c) [154]. Wang et al. fabricated a sword-like superhydrophobic ZnO film on wood substrate via an alkaline hydrothermal method [155]. Gao et al. used a low-temperature hydrothermal method to decorate $TiO_2$ nanoparticles on a wood substrate, and then achieved SHWSs by the chemical modification of low-surface-energy materials (i.e., OTS and FAS-17) [83,84]. Gao et al. obtained a robust superhydrophobic Ag-$TiO_2$ composite film on wood substrate via a two-step method by combining a hydrothermal method with silver mirror reaction [78].

The hydrothermal method, however, is not suitable for the formation of non-crystalline micro/nanostructured materials on wood substrates. In addition, this method requires special devices (i.e., Teflon-lined stainless-steel autoclave) with high vapor pressure to realize hierarchical roughness on wood substrates. Generally, this method has been extensively used in laboratory, while it still meets challenges of the scale-up preparation and safety in industry.

### 2.4. Dip-Coating

Dip-coating is a widely used method to construct superhydrophobic coatings because of its easy operation and high efficiency [34]. That is, the desired substrates are immersed into a solution of nanoparticles and/or low-surface-energy materials at a constant speed, and then experience a drying or curing process [146]. Due to rich surface hydrophilic functional groups, wood substrates are easily to react with chemical solutions of nanoparticles and/or low-surface-energy materials via the dip-coating method for the preparation of SHWSs. To achieve ideal SHWSs, the dip-coating process can be repeated for several times such that the thickness of superhydrophobic coatings can be controlled on wood substrates [3,158,159]. Similarly, the surface hierarchical roughness can also be changed by adjusting the concentrations of mixture solutions during a dip-coating process [160].

Kaewsaneha et al. obtained SHWSs by decorating wood substrates with alkyl ketene dimer (AKD) nanoparticles due to the reactions between AKD and the hydroxyls of cellulose via a dipping process (Figure 5a) [9]. Ding et al. achieved SHWSs by simply dipping the wood in the modifier solutions of hydrofluorosilicone oil (HFSO), tetramethyl tetravinyl cyclotetrasiloxane (V4) and hydrophobic SiO$_2$ (Figure 5b) and heating the resulting wood samples at 105 °C for 2 h [158]. Tu et al. applied silica/epoxy resin/FAS nanocomposites to wood substrate pre-coated with epoxy resin to realize SHWSs by a dipping process (Figure 5c) [159]. Janesch et al. prepared SHWSs by dip-coating spruce wood in a mixture of tung oil and natural beeswax, followed by the deposition of micronized sodium chloride particles (Figure 5d) [160]. Chang et al. successfully obtained SHWSs by simply dip-coating wood substrate with a suspension of hydrophobic silica nanoparticles and polydimethylsiloxane (Figure 5e) [91].

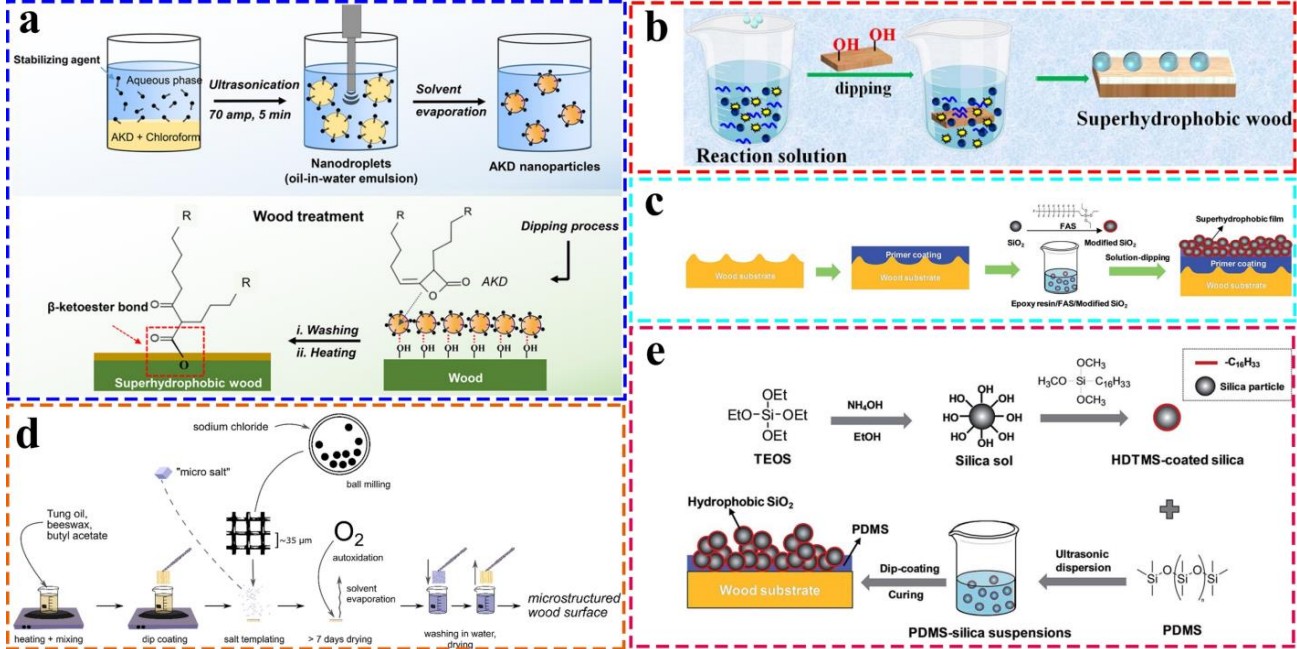

**Figure 5.** Schematic overview of preparing SHWSs by dip-coating: (**a**) Preparation of AKD nanoparticles via a nanoemulsion template method and the wood treatment process. (**b**) Wood modification with hydrofluorosilicone oil (HFSO), tetramethyl tetravinyl cyclotetrasiloxane (V4) and hydrophobic SiO$_2$. (**c**) Preparation of superhydrophobic coatings on wood with silica/epoxy resin/FAS. (**d**) Steps used to produce SHWSs. (**e**) The procedure to fabricate PDMS-silica hybrid superhydrophobic coatings on the wood substrate. Panel (**a**) reproduced with permission from ref. [9], copyright 2022, the authors. Panel (**b**) reproduced with permission from ref. [158], copyright 2022, the authors. Panel (**c**) reproduced with permission from ref. [159], copyright 2016, The Royal Society of Chemistry. Panel (**d**) reproduced with permission from ref. [160], copyright 2020, the authors. Panel (**e**) reproduced with permission from ref. [91], copyright 2016, The Royal Society of Chemistry.

To improve the durability of superhydrophobic coatings, either adding adhesive binders (i.e., epoxy resin) [22,159] or increasing repetition of dipping times [3,12,145,161] are carried out for preparing SHWSs during a dipping process. Furthermore, the drying or curing process is very important to preparing SHWSs by the dip-coating method, which can be controlled by the drying/curing temperature (i.e., 50 °C [92], 80 °C [93,162] and 103 °C [94]) and/or using vacuum-pressurized equipment [95,163]. In short, the dip-coating method can be scaled-up in real applications, which is suitable for the preparation of SHWSs.

## 2.5. Deposition

Deposition has been used for the preparation of SHWSs, including chemical bath deposition [164], solvothermal deposition [165], electroless deposition [166], chemical vapor deposition (CVD) [96], vapor deposition [97,98], plasma deposition [99], combustion deposition [61,100], thermal-driven deposition [167], reduction deposition [168,169] and plasma-enhanced chemical vapor deposition (PECVD) [170]. The above deposition methods are suitable for the preparation of SHWSs based on the properties (i.e., physicochemical and insulation properties) of wood substrates.

Zhang et al. successfully obtained a superhydrophobic $SiO_2$/PMHS coating on wood substrate by depositing $SiO_2$ nanoparticles and then hydrophobically treating with 1% PMHS modifier solution (Figure 6(a1,a2)) [101]. Wang et al. prepared SHWSs by immersing PDA-coated wood substrates in an electroless copper bath for 12 h, soaking the resulting wood substrates in octadecylamine ethanol solution at 30 °C for 24 h, and finally drying at 60 °C, respectively [166]. Wu et al. obtained SHWSs by depositing a polytetrafluoroethylene/microcapsule/zinc oxide (PTFE/MC/ZnO) dispersion on wood substrate, followed by the chemical modification of a mixture of stearic acid (4 wt.%) acidified with acetic acid in an ethanol solution [171]. Yao et al. achieved SHWSs with one-step solvothermal deposition of ZnO nanorod arrays on wood substrates by combining an immersion process, self-assembly and a hydrothermal method (Figure 6b) [165]. Liu et al. firstly treated wood substrate with epoxy resin acetone solution, deposited amino-functionalized silica particles on the epoxidized wood sample, and finally realized SHWSs with the self-assembly of an OTS monolayer (Figure 6c) [172]. Tan et al. realized superhydrophobic/superoleophilic wood flour via the deposition of ZnO nanoparticles and the subsequent chemical modification of octadecanoic acid (Figure 6d) [102]. Furthermore, the candle soot is also be used to decorate wood substrates by combustion. Li et al. fabricated SHWSs by spray-coating a mixture solution of pre-PDMS, n-hexane and cross-linking agent (the mass ratio is 10:10:1) on balsa wood, and depositing candle soot at a height of 2 cm over the flame [61].

To achieve SHWSs, deposition as an individual method is not enough to realize surface superhydrophobicity, and it is necessary to combine it with other techniques (i.e., spray-coating [61], immersion [166], self-assembly [8], reduction [173], dip-coating [95], magnetron sputtering [103] and chemical modification of low-surface-energy materials [171]). However, some of the deposition methods need special or expensive equipment (i.e., CVD and PECVD), which is not beneficial to the extensive and scaled-up preparations of SHWSs. Thus, the chosen and utilization of deposition methods for preparing SHWSs need to be investigated based on the actual demand and experimental conditions.

## 2.6. Sol-Gel Process

The sol-gel process is a simple, controllable and inexpensive method for preparing SHSs on various substrates [174]. The main purpose of the sol-gel method is to create surface roughness on the desired substrates (i.e., wood and fabrics). Generally, chemical solutions or colloidal solutions (sol) are used as precursors, and gels with three-dimensional networks can be obtained after a series of hydrolysis and polycondensation reactions [140,146,175]. Because of rich hydrophilic functional groups, superhydrophobic coatings are suitable to be formed on wood surfaces by using the sol-gel method and low-surface-energy modification [176].

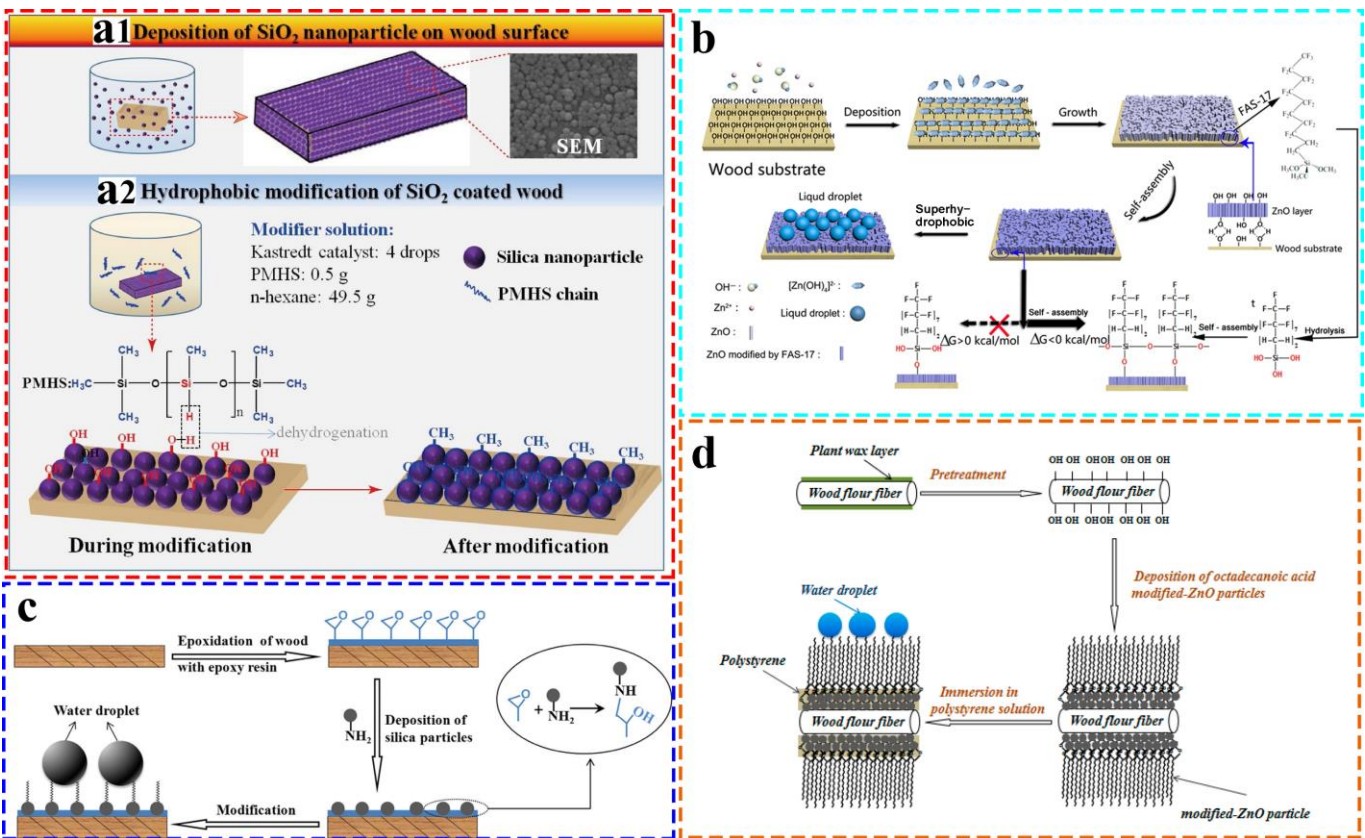

**Figure 6.** Schematic overview of preparing SHWSs by the deposition: (**a1**,**a2**) Fabrication of SiO$_2$/PMHS coatings on wood surface. (**b**) Preparation of the superamphiphobic ZNA-treated wood. (**c**) Preparation of superhydrophobic epoxy/SiO$_2$ coatings on wood substrates. (**d**) Preparation of superhydrophobic/superoleophilic wood flour. Panel (**a**) reproduced with permission from ref. [101], copyright 2019, Elsevier. Panel (**b**) reproduced with permission from ref. [165], copyright 2016, the authors. Panel (**c**) reproduced with permission from ref. [172], copyright 2015, De Gruyter. Panel (**d**) reproduced with permission from ref. [102], copyright 2019, the authors.

Tsvetkova et al. obtained SHWSs by coating wood substrates that were pre-coated by a primer (i.e., silica sol with boric acid or commercial reagent ethyl silicate-40) with a sol-gel@paint composition consisting of siloxane modifiers, tetraethoxysilane (TEOS) and aerosil R-972 (Figure 7a) [108]. Wang et al. successfully achieved SHWSs by depositing a film of silica nanoparticles on a wood substrate by using silica sol via a sol-gel process, followed by chemical modification with hydrophobic hexadecyltrimethoxysilane (HDTMS) and a drying process (Figure 7b) [109]. Xia et al. constructed a superhydrophobic coating of 1H,1H,2H,2H-perfluoroalkyltrichlorosilane (PFDS)-SiO$_2$ on wood substrate by using a sol-gel process and CVD (Figure 7c) [110]. Chang et al. fabricated superhydrophobic organic–inorganic coatings on wood substrates by using an inorganic precursor (i.e., tetraethoxysilane (TEOS)) and an organic modifier (i.e., hexadecyltrimethoxysilane (HDTMS)) via a sol-gel process (Figure 7d) [105].

To prepare SHWSs, the sol-gel method is usually combined with other techniques, such as the hydrothermal synthesis [17], the chemical modification of low-surface-energy materials [106], dipping [177] and electron beam curing [107]. As for the sol-gel methods, reactions need to experience a sol-gel process, which limit the resources of precursors or sols (i.e., SiO$_2$ sol and TiO$_2$ sol) for the preparation of SHWSs. Compared with other methods for preparing SHWSs, the sol-gel method can be scaled-up and applied on large-area surfaces, which facilitates extending its applications in our daily life and industry.

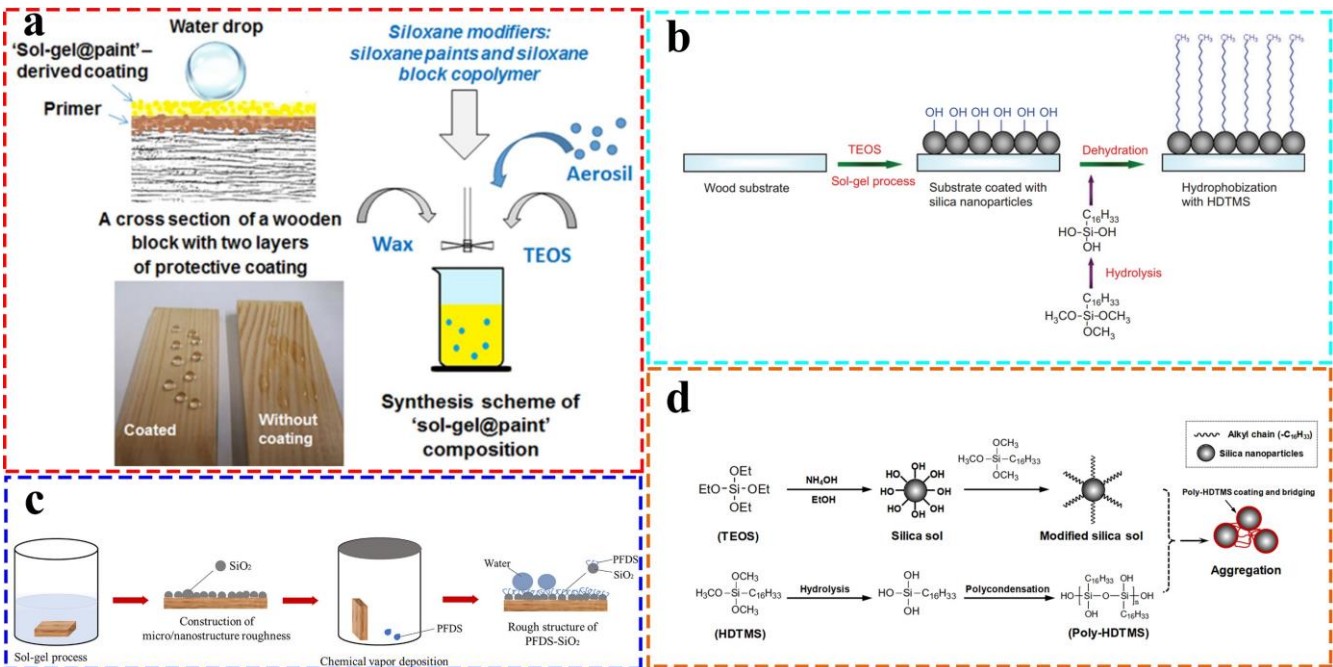

**Figure 7.** Schematic overview of preparing SHWSs by a sol-gel process: (**a**) Preparation of sol-gel@paint composition. (**b**) Formation of superhydrophobic wood surface based on silica sol and HDTMS. (**c**) The function mechanism of PFDS solution for obtaining superhydrophobic wood surface. (**d**) Hydrophobic modification of silica sols and aggregation of silica colloid particles. Panel (**a**) reproduced with permission from ref. [108], copyright 2019, Springer Science + Business Media, LLC, part of Springer Nature. Panel (**b**) reproduced with permission from ref. [109], copyright 2013, Walter de Gruyter Berlin Boston. Panel (**c**) reproduced with permission from ref. [110], copyright 2020, Elsevier. Panel (**d**) reproduced with permission from ref. [105], copyright 2015, NC State University.

*2.7. Other Methods*

Except for the above methods, there are also various other methods for preparing SHWSs, including coordination-driven multistep assembly [112], dehydrogenation-driven assembly [178], layer-by-layer assembly [15,179], self-assembling and retentions [111], brushing [12,180], drop-coating [181], impregnation [21,182], atom transfer radical polymerization (ATRP) [119], grafting [120,183], casting [121,122], solvothermal method [128], alkali-driven method [129], wet chemical process [184], multi-solvent continuous modification method [185], etching [186], carbonization [187], top-down strategy [188], soft lithography [130], in situ growth [107], hot-compression treatment [189], hot pulling technique [190], nanoimprint lithography [131], photochemical approach [191], spin-coating [192], silver mirror method [193], silanization [194,195], replication [132], etc.

Different assembly methods have been used for preparing SHWSs. Wang et al. prepared tannic-acid-FeIII-based superhydrophobic coatings on wood longitudinal surfaces with natural micro-grooved structures via a coordination-driven multistep assembly and chemical modification of octadecanethiol (Figure 8a) [112]. Lu et al. deposited $TiO_2$ (or $SiO_2$ [113]) nanoparticles on wood substrates by using a layer-by-layer assembly method and chemical modification of 1H,1H,2H,2H-perfluoroalkyltriethoxysilane (POTS) [114]. The assembly method is usually not enough to achieve surface superhydrophobicity, which needs to be further modified with low-surface-energy materials.



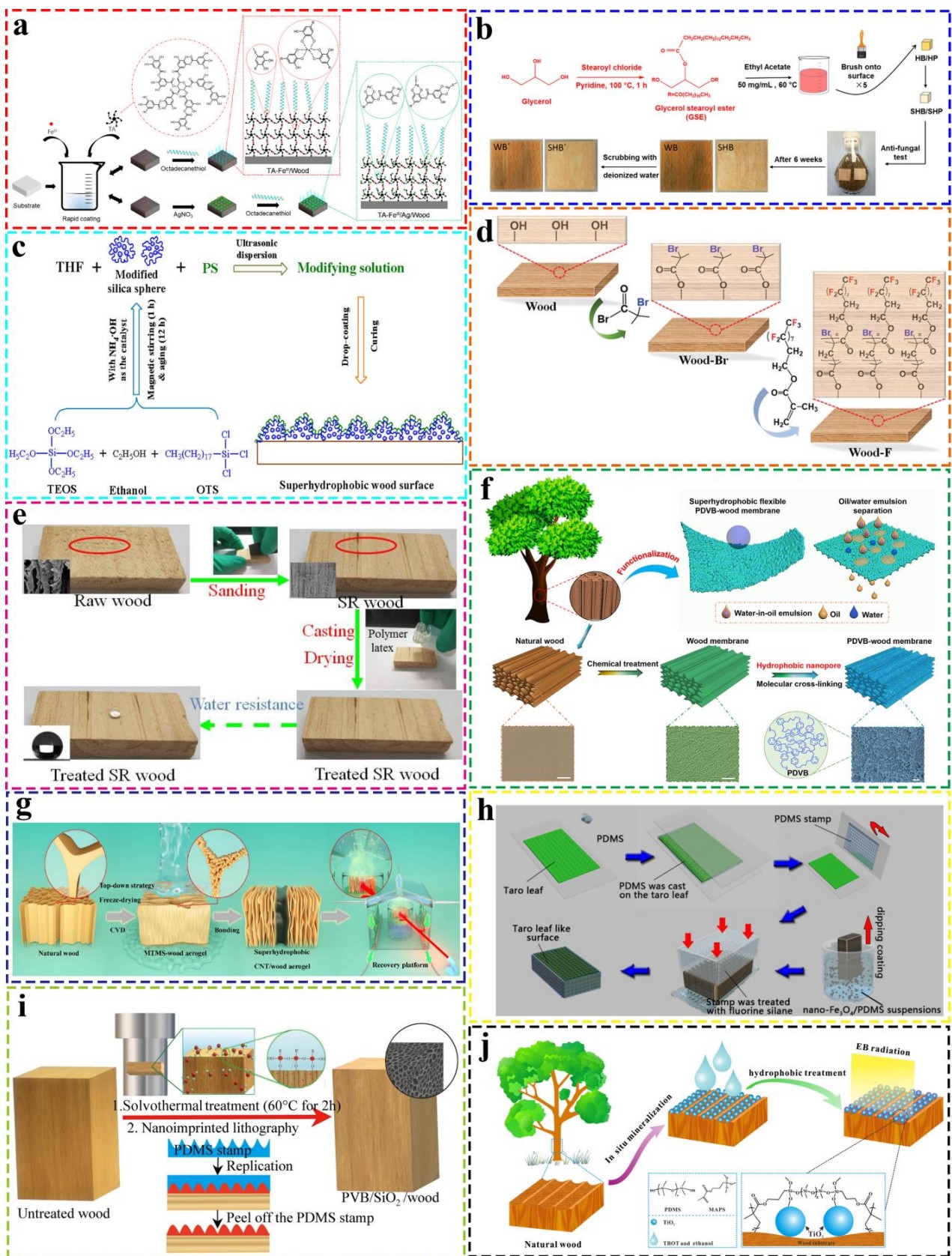

**Figure 8.** Other preparation methods for preparing SHWSs: (**a**) assembly, (**b**) brushing, (**c**) drop-coating, (**d**) atom transfer radical polymerization (ATRP), (**e**) casting, (**f**) solvothermal method, (**g**) top-down strategy, (**h**) soft lithography, (**i**) nanoimprint lithography, (**j**) in situ mineralization.

Panel (**a**) reproduced with permission from ref. [112], copyright 2017, the authors. Panel (**b**) reproduced with permission from ref. [16], copyright 2017, Springer Science Business Media B.V., part of Springer Nature. Panel (**c**) reproduced with permission from ref. [181], copyright 2013, The Society of Powder Technology Japan. Panel (**d**) reproduced with permission from ref. [119], copyright 2020, Walter de Gruyter GmbH, Berlin/Boston. Panel (**e**) reproduced with permission from ref. [121], copyright 2018, the authors. Panel (**f**) reproduced with permission from ref. [128], copyright 2022, the authors. Panel (**g**) reproduced with permission from ref. [188], copyright 2019, American Chemical Society. Panel (**h**) reproduced with permission from ref. [130], copyright 2017, Springer Science + Business Media New York. Panel (**i**) reproduced with permission from ref. [131], copyright 2019, the authors. Panel (**j**) reproduced with permission from ref. [107], copyright 2021, Springer Nature.

Compared with spray-coating, the brushing method is a very easy-operation approach to coat superhydrophobic coatings on wood substrates, which does not require expensive equipment or complex procedures [115]. Yao et al. fabricated SHWSs with two coating steps, which consists of preparation of the 1st layer by dip-coating in a cellulose stearoyl ester (CSE) solution and preparation of the 2nd layer by brushing a glycerol stearoyl ester (GSE) solution on the resulting wood samples (Figure 8b) [16]. However, superhydrophobic coatings on wood substrates are not determined by the methods (i.e., spraying, brushing and dipping), which sometimes only depends on the emulsion or suspension for preparing SHWSs [116].

The drop-coating method is a facile approach for preparing SHWSs, wherein wood substrates are coated by gradually dropping mixture solutions [23,118]. Wang et al. obtained SHWSs by dropping several drops of a mixture dispersion solution that consisted of polystyrene (PS), tetrahydrofuran (THF) and silica particles on wood substrate and consequently drying the sample at room temperature (Figure 8c) [181]. Liu et al. coated a poplar wooden substrate with a suspension of $SiO_2$ and polyvinyl alcohol (PVA), and then realized surface superhydrophobicity after treating with an octadecyltrichlorosilane (OTS) solution to obtain a self-assembled layer of the OTS monolayer [117]. As for the thickness of superhydrophobic coatings, it can be controlled by repeating the drop-coating procedure; the solvent should be completely evaporated before the next drop-coating procedure.

ATRP is an efficient method for grafting polymers on substrates, which can be accurately controlled. Wang et al. used wood-Br with active terminal bromine as the macro-initiator, and fabricated SHWSs by grafting poly(2-(perfluorooctyl)ethyl methacrylate) (PFOEMA) onto a wood substrate via ATRP (Figure 8d) [119]. Grafting as a facile approach is also used to decorate wood substrates with long-chain alkyls to realize surface superhydrophobicity. Wang et al. successfully prepared highly hydrophobic wood substrates by grafting long-chain alkyl groups onto wood cell walls via urethane linkage (or ester linkage [120]) [183].

Casting is a simple and low-cost approach for constructing SHSs on wood substrates. Shen et al. drop-casted polymer latexes onto sanded wood surfaces, and obtained SHWSs by drying the above polymer latexes at 30 °C (Figure 8e) [121]. Similarly, Wang et al. casted a fluoroalkylsilane/silica composite suspension onto sanded wood substrate, and achieved SHWSs by drying at 120 °C for 2 h [122]. Compared with the drop-coating method, the casting method has a reconstructing process for the polymer coatings by drying.

Impregnation is a process of permeating or infusing chemicals on the chosen substrates, which is suitable for preparing durable superhydrophobic coatings on wood substrates with natural roughness (or porous structures) [21,123]. For example, Yang et al. prepared durable SHWSs by using a two-step impregnation consisting of chemical modification with natural maleic rosin and subsequent coating with $TiO_2$ nanoparticles to increase surface roughness [124]. Of course, the impregnation approach can be improved by the assistance of vacuum with pressure (i.e., 0.01 MPa [125], 0.5 MPa [125] and −0.1 MPa [13]). Liu et al. realized SHWSs by the impregnation of a silica/silicone oil complex emulsion into wood substrates with the help of vacuum (i.e., 0.01 MPa for 1 h and 0.5 MPa for 1 h) and drying with different temperatures (i.e., 60 °C for 12 h, elevated to 80 °C for 12 h, and

elevated to 103 °C for drying) [125]. Furthermore, the impregnation method can combine with other methods (i.e., hydrothermal method [80], drying [126] and low-surface-energy modification [127]) for preparing SHWSs.

Some solvent-based methods have been introduced for preparing SHWSs, such as the solvothermal method [128], alkali-driven method [129], wet chemical process [184] and multi-solvent continuous modification method [185]. For example, Cai et al. prepared superhydrophobic (polydivinylbenzene) PDVB-wood membrane by implementing PDVB onto wood substrate via a solvothermal method (Figure 8f) [128]. Jia et al. utilized an alkali-driven method by using $SiO_2$ nanoparticles and vinyltriethoxysilane (VTES) [129]. Wang et al. fabricated a lamellar superhydrophobic ZnO coating on wood substrate by using a wet chemical process followed by modification of stearic acid [184]. Yang et al. fabricated a polyethylene glycol (PEG)-functionalized $SiO_2$/poly(vinylalcohol) (PVA)/poly(acrylic acid) (PAA)/fluoropolymer hybrid by using a multi-solvent continuous modification method, including chemical reaction, solution self-assembly and impregnation [185]. To some extent, the solvent-based methods may be similar to the immersion method, while the driving force for preparing SHWSs is different.

Silanization is a very facile approach to modifying wood substrates. Yin et al. modified birch wood substrate by the silanization of silicone nanofilaments and 1H,1H,2H,2H-Perfluorodecyltrichlorosilane (PFDTS) in a sealed desiccator [195]. Zhang et al. fabricated superhydrophobic coatings on various substrates (i.e., plywood, filter paper, aluminum, cotton fabric and plastic) by a single-step, stoichiometrically controlled reaction of long-chain organosilanes with water [194]. Zhu et al. fabricated wood aerogels by using a top-down strategy to remove lignin and hemicellulose, followed by freeze-drying (Figure 8g) [188]. They successfully obtained SHWSs after the treatment of methyltrimethoxysilane (MTMS) in an in-house reaction chamber via the CVD method [188]. To achieve surface superhydrophobicity, the silanization method as a single approach is not enough for many smooth substrates (i.e., glass, metals and polymers), while it is suitable for preparing SHSs on wood substrates with natural hierarchical surface roughness.

Bio-inspired from nature, template methods are also used for preparing SHWSs, including soft lithography, nanoimprint lithography and replication. Chen et al. obtained superhydrophobic taro-leaf-like film-decorated wood surfaces by using the PDMS stamp (as being peeled from the taro leaf) and PDMS/nano-$Fe_3O_4$ composite suspensions via soft lithography (Figure 8h) [130]. Yang et al. realized a superhydrophobic PVB/$SiO_2$-coating-decorated wood substrate by using a one-step solvothermal method and a nanoimprint lithography method (Figure 8i) [131]. Yang et al. prepared canna-leaf-like micro/nanostructures on a wood substrate by using PDMS as a template and replicating polyvinyl butyral (PVB)/$SiO_2$ coatings on wood substrate [132]. Although the template method is a well-controlled approach to preparing SHWSs, it is still not suitable for large-scale production of SHWSs due to the special and complex procedures required.

In addition to this, there are some interesting methods preparing SHWSs. For example, Li et al. reported that a robust superhydrophobic coating can be successfully achieved on wood substrate through in situ mineralization and polymerization (Figure 8j) [107]. Wu et al. used a silver mirror method to realize in situ growth of silver nanoparticles on wood substrate followed by the chemical modification of a mixture solution of stearic acid (4 wt.%) and ethanol [193]. In summary, the above methods of preparing SHWSs have own unique characteristics, and they can be chosen based on cost and experimental conditions. In addition, the practical applications of wood materials should be considered; this is because SHWSs need to have different physiochemistry and mechanical properties in different environments and different practical applications. Furthermore, to enhance surface superhydrophobicity, multiple of the above methods can be combined such that SHWSs can be successfully obtained with excellent properties in different experimental conditions.

## 3. Applications of SHWSs

Due to excellent surface superhydrophobicity, SHWSs has been extensively utilized in various fields, including anti-fungi [16], anti-bacteria [119], oil/water separation [102], fire resistance [133], anti-UV irradiation [156], photo-response [137], electromagnetic interference (EMI) shielding [79], anti-icing [177] and wood-based devices [147]. In this section, we summarize recent applications of SHWSs, and discuss challenges for the versatile application of SHWSs.

### 3.1. Anti-Fungi and Anti-Bacteria

When exposed to favorable environments (i.e., moisture, sufficient air and fat), hydrophilic wood surfaces can be easily degraded and damaged due to the attacks of fungi and bacteria [16]. Traditionally, inorganic waterborne preservatives (i.e., chromated copper arsenate) [21] and organic mold inhibitors (i.e., 4,5-dichloro-2-octyl-isothiazolone) [196] have been used to protect wood surfaces. However, they suffer from harms to human beings and environments, degradation and decomposition [21,197]. Recently, an alternative strategy is to introduce SHSs, which can serve as water barriers and thus prevent SHWSs from being permeated by moisture [16]. Up to now, many SHWSs have been used in the fields of anti-fungi [16,21] and anti-bacteria [19,78,119,193].

For example, Yao et al. prepared SHWSs by dip-coating and brush-coating, and found that SHWSs can thoroughly prevent fungal attachment to wood surfaces compared to hydrophobic or hydrophilic wood surfaces [16]. This is because SHWSs work as water barriers and can isolate moisture with wood structures, thus limiting water resources for the growth of fungi and bacteria. Another reason is that SHWSs have very low adhesion, which makes the attachment of fungi and bacteria on SHWSs difficult. Lu et al. carried out mold-proof experiments with rubberwood and superhydrophobic rubberwood by using four kinds of molds (i.e., Botryodiplodia theobromae, Trichoderma viride, Penicillium citrinum and Aspergillus niger), and found that superhydrophobic rubberwood shows much better anti-mold performance than that of rubberwood (Figure 9(a1–a8)) [21]. Compared with rubberwood, the attachment of molds onto superhydrophobic rubberwood is difficult (Figure 9(a5–a8)). Wang et al. obtained SHWSs by grafting poly(2-(perfluorooctyl)ethyl methacrylate) onto wood by atom transfer radical polymerization [119]. It can be found in Figure 9(b1–b6) that the color of the pristine wood changes after the cultivation of Aspergillus niger, and the hyphae gradually penetrates into wood [119]. As for SHWSs, no mold colony can be observed even after 15 days of cultivation (Figure 9(c1,c2)) [119]. Duan et al. prepared SHWSs by self-polymerization of dopamine, chemical deposition of Cu nanoparticles and hydrophobic modification of fluorosilane, and the SHWSs possessed excellent antibacterial performance for killing *E. coli* and *S. aureus* (Figure 9(d1–d6)) [19]. Gao et al. fabricated superhydrophobic Ag/TiO$_2$-coated wood by hydrothermal synthesis, silver mirror reaction and chemical modification [78]. They found that TiO$_2$-treated wood does not show any antibacterial activity, while Ag/TiO$_2$-coated wood can kill all the bacteria under and around them [78].

The reason why SHWSs can effectively prevent fungi and bacteria is that SHWSs possess very low adhesion for both fungi and bacteria, and also show excellent water repellency to stop water moisture. These two advantages inhibit the living environments of fungi and bacteria, and thus can prevent the growing of fungi and bacteria for a long time. In other words, the anti-fungi and anti-bacteria properties of SHWSs can be maintained as long as the surface superhydrophobicity of SHWSs exists. As time elapses, however, the surface superhydrophobicity of SHWSs cannot be guaranteed under extremely humid environments towards various fungi and bacteria. A realistic roadmap towards protecting wood materials is to introduce SHWSs as well as the regular maintenance of SHWSs. In a word, self-cleaning and anti-fouling properties of wood surfaces can prevent wood materials from being molded, which will extend their real outdoor practical applications.

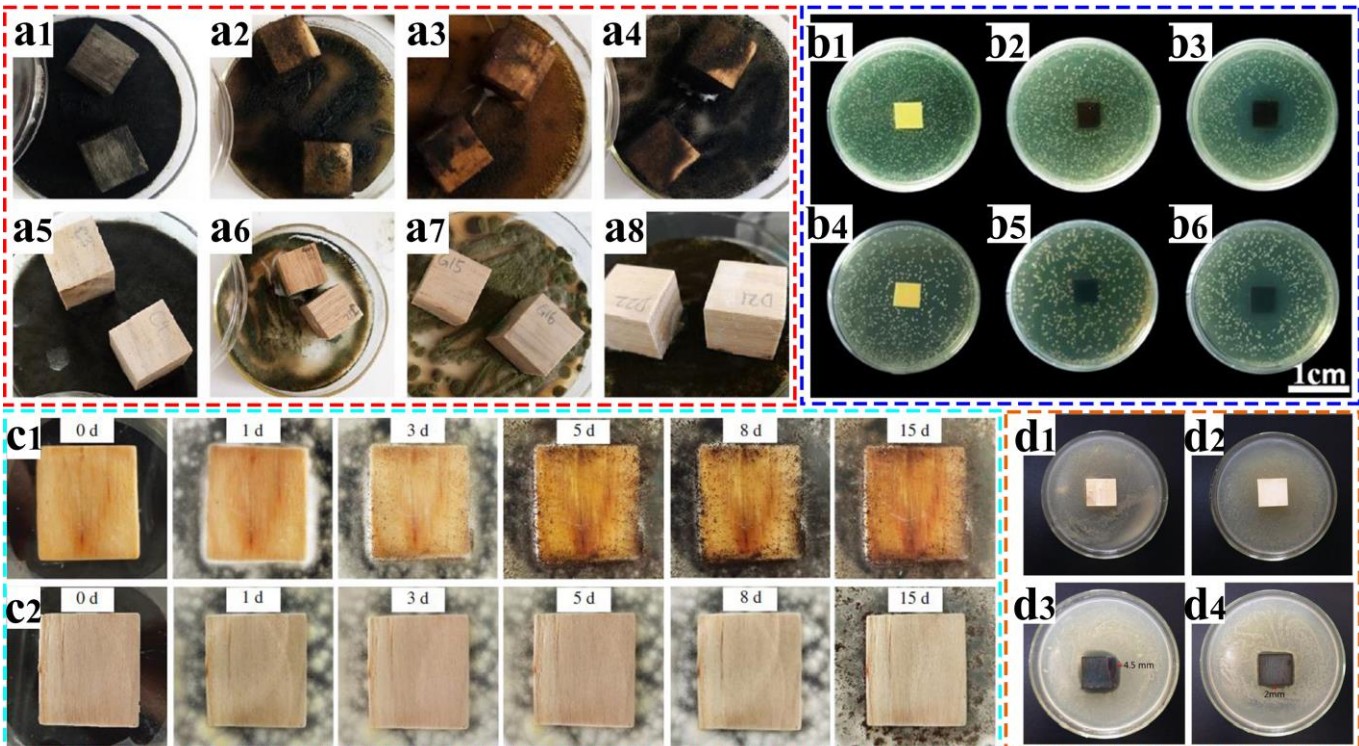

**Figure 9.** Photographs of mold-proof specimen: (**a1**–**a4**) rubberwood, (**a5**–**a8**) superhydrophobic rubberwood. Antibacterial activities of (**b1**) wood, (**b2**) wood@PDA and (**b3**) superhydrophobic wood in *E. coli*, and (**b4**) wood, (**b5**) wood@PDA and (**b6**) superhydrophobic wood in *S. aureus*. Images for anti-mold activity against Aspergillus niger: (**c1**) pristine wood, (**c2**) superhydrophobic wood. Antibacterial activities of (**d1**,**d2**) the hydrophobized $TiO_2$-treated wood in Escherichia coli and Staphylococcus aureus, and (**d3**,**d4**) the hydrophobized $Ag/TiO_2$-coated wood in *E. coli* and *S. aureus*. Panel (**a**) reproduced with permission from ref. [21], copyright 2019, NC State University. Panel (**b**) reproduced with permission from [19], copyright 2020, Elsevier. Panel (**c**) reproduced with permission from ref. [119], copyright 2020, De Gruyter. Panel (**d**) reproduced with permission from [78], copyright 2016, Elsevier.

### 3.2. Oil/Water Separation

Oceanic oil spills and the discharge of oily wastewaters have caused significant threats to the ecological environment and human health [32,198,199]. To mitigate this issue, traditional methods have been utilized to remove toxic chemicals in oily wastewaters, including in situ burning, air flotation, centrifugation, chemical dispersion, and bioremediation [24,32]. These methods, however, suffer from complex operations, high cost, low separation efficiency, and secondary pollution [198,199]. Thus, adsorption and separation materials draw much attention for dealing with oily wastewaters. Recently, many materials have be used in the field of oil/water separation, such as papers [24], polymers (i.e., polyester fabrics) [200], metal meshes [201], cottons [202] and wood [203]. Among them, the sustainable superhydrophobic wood material with porous structures is an excellent candidate for oil/water separation or oil recovery (See Table 2) [203].

Latthe et al. utilized low-cost sawdust and polystyrene to prepare superhydrophobic pellets that possess micro-voids less than 100 μm [25]. These superhydrophobic pellets show excellent oil/water separation efficiency higher than 90% as well as separation cycles around 30 for oils and organic solvents (Figure 10(a1–a4)) [25]. Chen et al. obtained a superhydrophobic delignified wood material by coating $TiO_2$ and PDMS, and this superhydrophobic wood can easily absorb underwater oils (Figure 10b) [162]. This superhydrophobic wood shows a high permeation flux of up to 6111 L m$^{-2}$ h$^{-1}$ and a separation efficiency of up to 93.4% [162]. Wang et al. prepared superhydrophobic wood

aerogel/PDMS composite by partially removing hemicellulose and lignin and treating with PDMS [135]. This prepared wood aerogel/PDMS membrane material can efficiently separate oil/water mixtures with high separation efficiency (99.5%) and flux (around $2.25 \times 104$ L m$^{-2}$ h$^{-1}$) by only using the driving force of gravity (Figure 10(c1,c2)) [135]. Ma et al. achieved a bio-based fireproof and superhydrophobic wood template for oil/water separation via a layer-by-layer assembly technique, and this superhydrophobic wood showed excellent oil/water properties (>97%), self-cleaning capability, and mechanical durability (Figure 10(f1–f4,g1–g6)) [179]. Although the oil/water separation performance of SHWSs is not as good as that of sponges [199], SHWSs are still good candidates because of their high efficiency and recyclability (Table 2).

**Table 2.** Oil/water separation of SHWSs.

| Modifying Materials | Adsorbed Materials | Adsorption Capacity (g·g$^{-1}$) | Separation Efficiency (%) | Permeation Flux (L·m$^{-2}$·h$^{-1}$) | Cycles (Times) | Ref. |
|---|---|---|---|---|---|---|
| Acetic acid/NaClO$_2$ | Toluene | 14.25 | 94 | 32.8 | 100 | [2] |
| Polystyrene | Hexane | - | >98 | - | 36 | [25] |
| PDMS | Trichloromethane | 20 | 99.5 | $2.25 \times 10^4$ | 10 | [135] |
| TEOS/PDMS | Dichloromethane | - | 98.5 | $1.3 \times 10^3$ | 3 | [104] |
| NaClO$_2$ | Dichloromethane | 37 | 88.6 | - | 10 | [138] |
| STA/PF/Cu$_2$O | Dichloromethane | 3.2 | 94 | - | 30 | [55] |
| POTS/SiO$_2$ | Emulsions | - | >99 | 710 | 10 | [71] |
| PDMS/TiO$_2$ | Silicone oil | 2.7 | 93.4 | 6111 | 20 | [162] |
| ZnO/octadecanoic acid | Engine oil | 20.81 | >98 | - | 10 | [102] |
| APTES/SiO$_2$ | Chloroform | - | 97.5 | - | - | [179] |
| PDVB | Toluene emulsions | - | 99.98 | 8829.4 | 20 | [128] |
| MTMS | Olive oil | 23.1 | - | - | 10 | [188] |
| Cu(OH)$_2$ | Emulsions | - | >98 | 11 | 6 | [123] |
| PSP | Emulsions | - | >99 | 4392 | 12 | [204] |
| Na$_3$(Cu$_2$(CO$_3$)$_3$OH)·4H$_2$O | Dichloromethane | 5 | 90 | - | 11 | [205] |

As for SHWSs, natural porous structures are not enough for high oil adsorption capacity, while superhydrophobic wood aerogels with delignified structures may be promising for oil/water separation. Actually, the challenges for oil/water separation are mainly the separation of viscous oils and oil/water emulsions. One approach for designing SHWSs with excellent oil/water separation properties is to decorate delignified wood surfaces with various desired nanoparticles and thus achieve suitable hierarchical surface roughness. Another approach is to introduce the photo-thermal effect to enhance the oil/water separation by decorating with light-absorbing nanoparticles, which facilitates to decrease the viscosity of oil/water mixtures (or emulsions) and thereby improve the separation efficiency.

### 3.3. Fire Resistance

Although wood materials have many advantages (i.e., high strength, easy machinability, aesthetic characteristics) for indoor and outdoor construction and decorations, the combustion of wood materials is an unavoidable problem for the safe utilization of wood materials [5,17,133,179]. With wood materials have combustion characteristics, a possible solution is to endow wood materials with fire resistance. One traditional method is to treat wood materials with fire retardants; however, these fire retardants are harmful to the environments. With the assistance of adhesives, durable flame retardant coatings have been utilized for improving the fire resistance of wood materials [206]. Among various retardant coatings, superhydrophobic coatings consisting of metal or inorganic oxides (i.e., ZnO [5], Mg-Al-layered double-hydroxide [17] and SiO$_2$ [133,179]) have been successfully introduced onto wood surfaces, showing excellent fire-resistance properties. The fire-resistance property of SHWSs is possibly due to the combination of the fire resistance of metal oxides (or inorganic oxides) and the superhydrophobic coatings of wood materials acting as a protective layer.

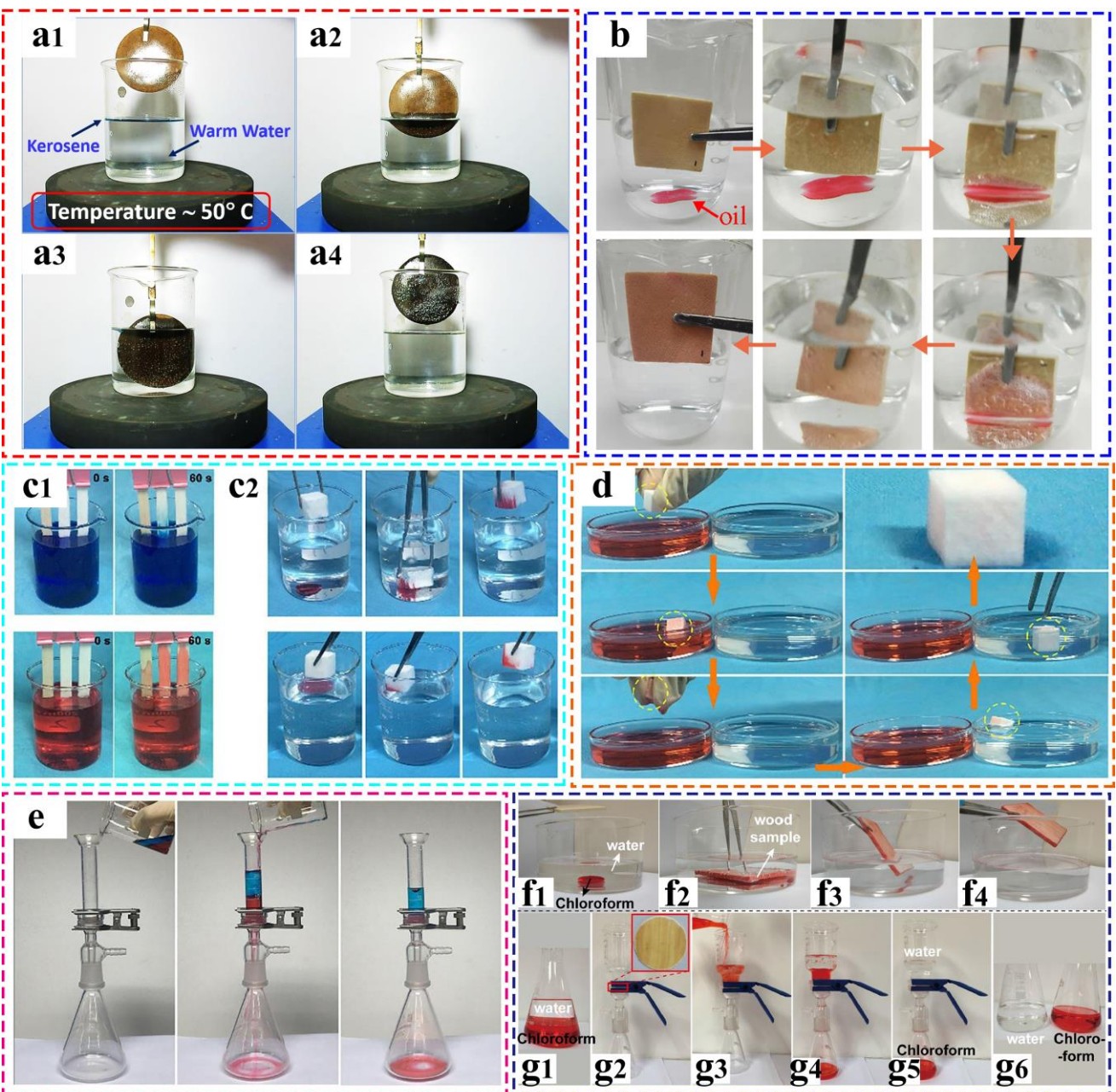

**Figure 10.** (**a1**–**a4**) Oil separation from the mixture of warm water (~50 °C) and kerosene oil by a superhydrophobic pellet. (**b**) Underwater absorption of PDMS@TiO$_2$ wood for dichloromethane. (**c1**) Capillary absorption of water (dyed to blue) and n-hexane (dyed to red) in balsa wood, wood aerogel and wood aerogel/PDMS composite. (**c2**) Removal of trichloromethane from water and n-hexane on the water surface. (**d**) Shape-memory function for the wood aerogel/PDMS composite. (**e**) Oil/water separation by using the wood aerogel/PDMS membrane. (**f1**–**f4**) The separation of chloroform from the water. (**g1**–**g6**) The procedure of the oil/water separation. Panel (**a**) reproduced with permission from ref. [25], copyright 2020, Elsevier. Panel (**b**) reproduced with permission from ref. [162], copyright 2022, Elsevier. Panel (**c**–**e**) reproduced with permission from ref. [135], copyright 2019, Elsevier. Panel (**f**,**g**) reproduced with permission from ref. [179], copyright 2021, Springer Science + Business Media, LLC, part of Springer Nature.

Kong et al. deposited highly dense and uniform ZnO arrays on the wood surface and achieved a superhydrophobic wood surface with a water contact angle of ~154° [5]. They carried out flame retardancy tests for pristine wood (Figure 11(a1)) and ZnO-nanorod-

coated wood (Figure 11(a2,a3)), and the experimental results showed that ZnO nanorod arrays can be used as a thermal protective layer for fire resistance [5]. Jia et al. prepared superhydrophobic wood surfaces by immersing wood surfaces into a sealed vessel with a mixture of massive epoxy and $SiO_2$ nanoparticles [133]. This superhydrophobic wood took 50 s for the surface to be entirely covered by flame, and 2.5 times longer than the untreated wood surface (Figure 11(b1,b2)) [133]. Ma et al. verified flame retardancy by using UL-94 tests, and found that superhydrophobic wood surfaces can easily pass the UL-94 test (Figure 11c) [179]. Guo et al. studied the combustion properties of superhydrophobic Mg-Al LDH-coated wood by using LOI and CONE calorimetry testing, and found that the LOI value can increase from 18.9% ± 1.3% to 39.1% ± 2.7% compared with the untreated wood [17]. Based on the above examples, SHWSs can be considered as good candidates for the application of fire resistance.

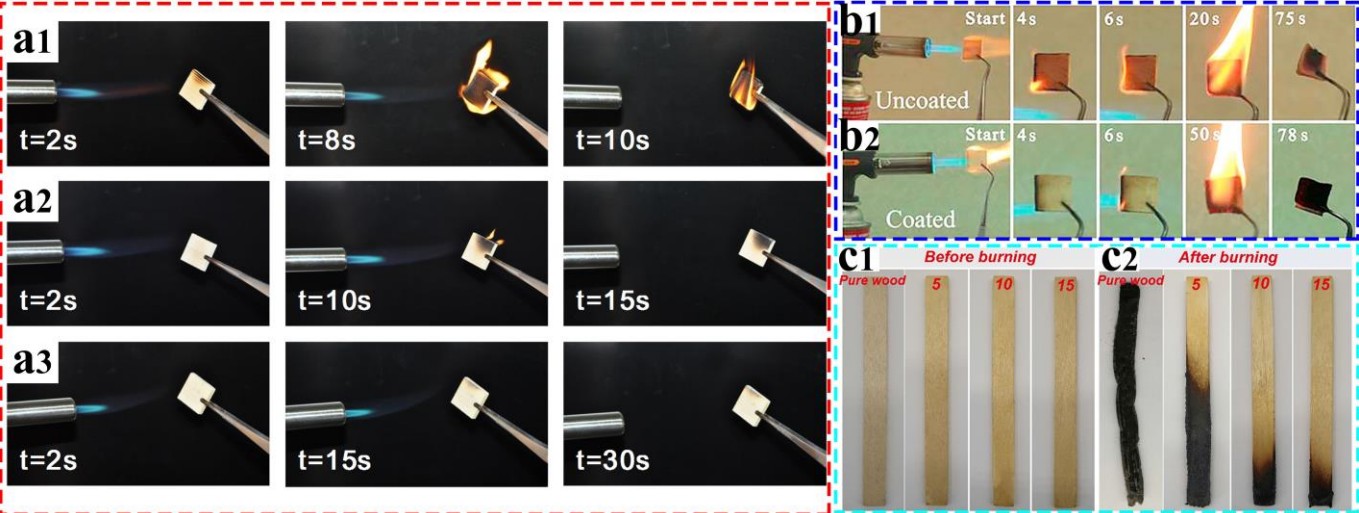

**Figure 11.** Flame retardancy tests for (**a1**) pristine wood and ((**a2**), 0.02 M; (**a3**), 0.1 M) ZnO nanorod-coated wood. The combustion process of (**b1**) the untreated wood and (**b2**) superhydrophobic wood. The digital photographs of wood samples (**c1**) before and (**c2**) after UL-94 tests. Panel (**a1**–**a3**) reproduced with permission from ref. [5], copyright 2017, Elsevier. Panel (**b1**,**b2**) reproduced with permission from ref. [133], copyright 2018, Elsevier. Panel (**c1**,**c2**) reproduced with permission from ref. [179], copyright 2021, Springer Science + Business Media, LLC, part of Springer Nature.

To achieve the property of fire resistance, the design of wood materials may focus on properties of inorganic coatings when preparing SHWSs, considering the thickness of coatings, thermal stability, flammability and smoke production. To be realistic, SHWSs can prevent combustion for a certain time, while they cannot constantly stop the flame for a long time as time elapses and flame temperature increases. All in all, superhydrophobic coatings on wood materials can be used as retardant coatings, and the fire resistance property of SHWSs is enough for wood materials as applied in some indoor or outdoor practical applications.

### 3.4. Anti-UV Irradiation

When wood materials are exposed to external environments, anti-UV irradiation should be considered such that the longevity of wood surfaces can be estimated. Generally, anti-UV irradiation properties are investigated by monitoring the color changes of pristine wood and SHWSs over a period of time [152]. For example, Jnido et al. found that the color change in uncoated wood is clearly recognizable, while the color change in the polyester/$TiO_2$-coated wood is not immediately visible [152]. Li et al. showed that the color of the pristine wood became noticeably darker and yellow after 18 days of UV irradiation, while no visible color change on superhydrophobic wood surfaces was found [152]. By

introducing nanomaterials (i.e., TiO$_2$ [107,152,176], ZnO [165] and CoFe$_2$O$_4$ [156]) with anti-UV properties, wood materials can be endowed with anti-UV properties and, thus, the utilization of wood materials can be enhanced in outdoor fields.

The color changes before and after UV irradiation are measured in accelerated aging tests to evaluate the anti-UV property of the pristine wood and SHWSs [156]. Usually, the change tendency of the chromaticity parameters (Δa* (a tendency to turn reddish), Δb* (a tendency to turn yellowish)), the lightness (ΔL*) and the overall color change (ΔE*) of the pristine wood and SHWSs is characterized. For example, Gan et al. found that the value of Δa* for the pristine wood and SHWSs is similar, while the value of Δa* for SHWSs is only 1/10 that of the untreated wood as shown in Figure 12(a1–a4) [156]. Wang et al. found that the nano-TiO$_2$ treatment significantly improves the photostability of wood surfaces [176]. Yao et al. also found that the superhydrophobic ZNA-treated wood shows superior UV resistance compared to that of the pristine wood Figure 12(b1–b4) [165]. In addition to this, the color fastness and color stability of wood materials can be also investigated by the characterization of their anti-UV properties. For example, Wang et al. studied color fastness enhancement of dyed wood with a Si-sol@PDMS-based superhydrophobic coating [95]. Tuong et al. prepared superhydrophobic epoxy@ZnO coated wood, and found that the color stability can be improved by around 50% compared with that of the uncoated wood [74]. Thus, the superhydrophobic coatings are suitable to be utilized for the protection of wood materials for UV resistance.

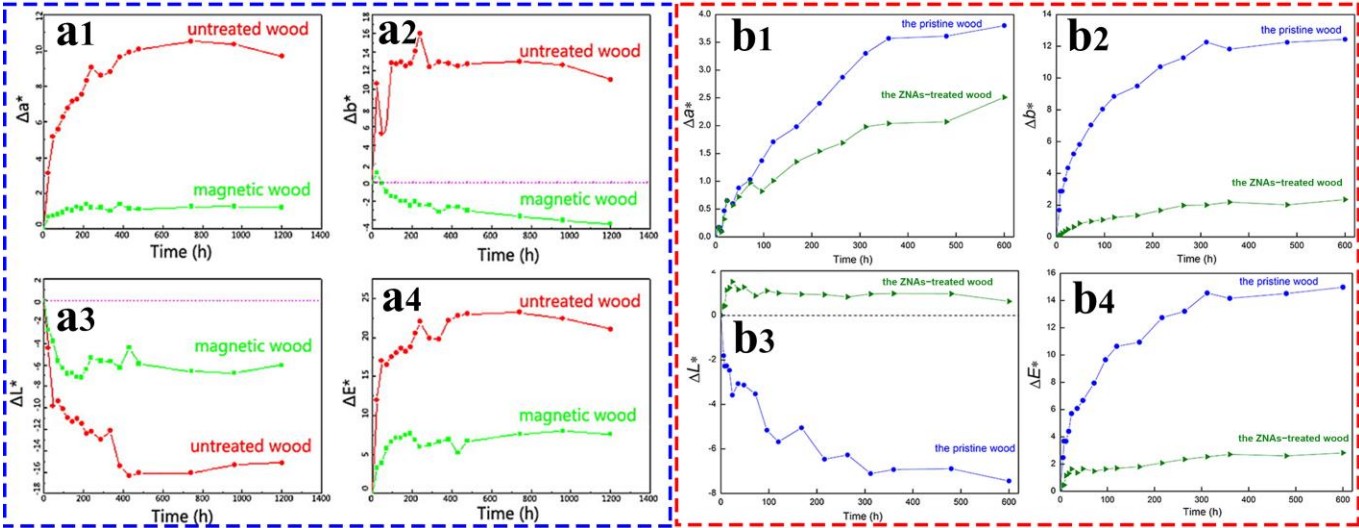

**Figure 12.** Change tendency of Δa*, Δb*, ΔL* and ΔE* of the untreated wood and the magnetic wood. Panel (**a1**–**a4**) reproduced with permission from ref. [156], copyright 2015, Elsevier. Panel (**b1**–**b4**) reproduced with permission from ref. [165], copyright 2016, the authors.

In short, as for outdoor applications of wood materials, the anti-UV property is one of most important factors that determine the application of wood materials. Solar irradiation (i.e., UV irradiation) can be easily absorbed by the component lignin in wood materials, thereby gradually degrading wood materials as time elapses. To enhance anti-UV properties, delignified wood materials may be a good choice when used as wood substrates when preparing SHWSs. Overall, the introduction of superhydrophobic coatings with anti-UV properties is also a realistic approach towards protecting wood materials in outdoor environments.

### 3.5. EMI Shielding

As the use of portable electronic devices increases, massive electromagnetic wave pollutions significantly influence living environments, which has a detrimental impact on human health [207]. To mitigate this issue, EMI shielding and electromagnetic (EM) wave

absorption materials have been developed such that public health can be protected. When preparing SHWSs, superhydrophobic coatings are suitable for use as a protective layer against EMI interference.

Metal (or oxides) (i.e., copper [48], $MnFe_2O_4$ [79] and $CoFe_2O_4$ [22])-based superhydrophobic coatings have been decorated onto wood materials. Xing et al. tested electromagnetic radiation intensity of three types of wood by using a cell phone (Figure 13(a1)) [48]. They found that the electromagnetic intensity of ordinary wood, coppered wood and superhydrophobic coppered wood is 1.686 µT, 0.446 µT and 0.358 µT (Figure 13(a2)), respectively [48]. Wang et al. showed that the absorption bandwidth of the FMW can be significantly improved compared with the wood (Figure 13(b1,b2)) [79]. Gan et al. found that the reflection loss of the coated wood composites is much lower than that of the pristine wood [22]. Thus, the EMI interference and/or EM wave absorption can be realized on wood materials.

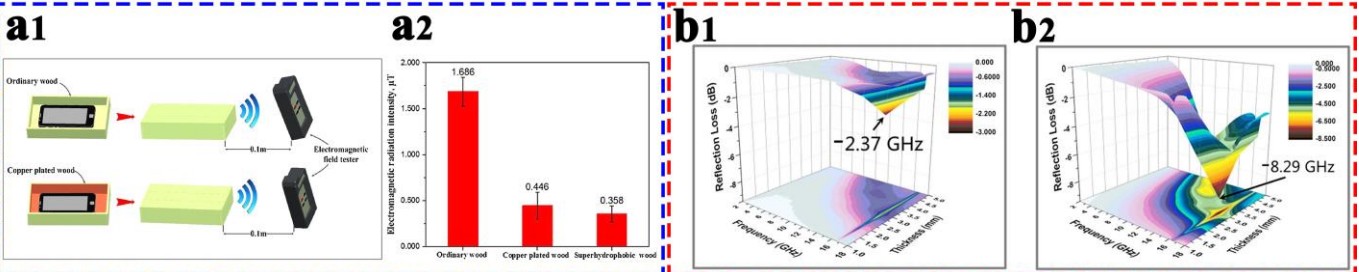

**Figure 13.** (**a1**) The process of testing electromagnetic radiation intensity. (**a2**) Electromagnetic radiation intensity of different types of wood. Frequency dependence of the reflection loss for (**b1**) the wood and (**b2**) the FMW by three-dimensional and color-filling patterns in the frequency range of 2–18 GHz. Panel (**a1**,**a2**) reproduced with permission from ref. [48], copyright 2017, Elsevier. Panel (**b1**,**b2**) reproduced with permission from ref. [79], Copyright 2016, the authors.

Now that the widespread use of portable electronic vehicles and devices are unavoidable in modern life, the mitigation of EM wave pollutions has become a huge and realistic challenge. As for indoor and outdoor construction and decorations, the introduction of SHWSs can be considered such that wood materials can simultaneously possess many excellent performances (i.e., self-cleaning and water repellency) as well as the abilities of EMI shielding and EM wave absorption.

### 3.6. Photocatalytic Performance

Self-cleaning is an excellent property of SHSs, which facilitates automatically removing contaminants (i.e., dust, powder, wastewater and organic contaminants). Inorganic contaminants can be easily removed by SHSs, while organic contaminants may stay on SHSs and thus gradually destroy the superhydrophobicity of SHSs as time elapses. If SHSs possess the self-cleaning property as well as photocatalytic performance, the self-cleaning property of SHSs can be definitely enhanced in various different environments.

Recently, SHWSs have been developed to have a good photocatalytic functionality in the degradation of organic contaminants. For example, Wang et al. prepared thermally induced responsive superhydrophobic wood, and studied the performance of the TRS-wood on the degradation of oleic acid (Figure 14(a1–a4)) [208]. They found that the TRS-wood shows a degradation efficiency of above 90% (Figure 14(a3)) [208]. Gao et al. obtained superhydrophobic $Ag/TiO_2$-coated wood, and investigated the photodegradation of phenol [78]. Jia et al. tested the degradation efficiencies of three catalytic samples (i.e., the untreated wood, Bi-wood, and Bi&F-wood), and the photocatalytic degradation efficiency was almost 100% for the Bi&F-wood after 80 min UV irradiation (Figure 14(b1–b6)) [139]. Xia et al. prepared superhydrophobic $STA@PF@Cu_2O$-coated balsa, and found that the concentrations of methylene blue declined substantially to 99.8% when exposed to visible light after 320 min [55]. Chen et al. prepared superhydrophobic $PDMS@TiO_2$ wood for pho-

tocatalytic degradation [162]. They found that the $TiO_2$ loading and the surface area of the samples are both important factors affecting the photocatalytic degradation efficiency [162]. When testing the photocatalytic degradation efficiency, they used crushed PDMS@$TiO_2$ wood to increase surface contact area and thus improve the photocatalytic degradation efficiency between the $TiO_2$ coating and UV light [162]. Based on above discussions, SHWSs can be endowed the photocatalytic performance, and thus the applications of SHWSs can be extended.

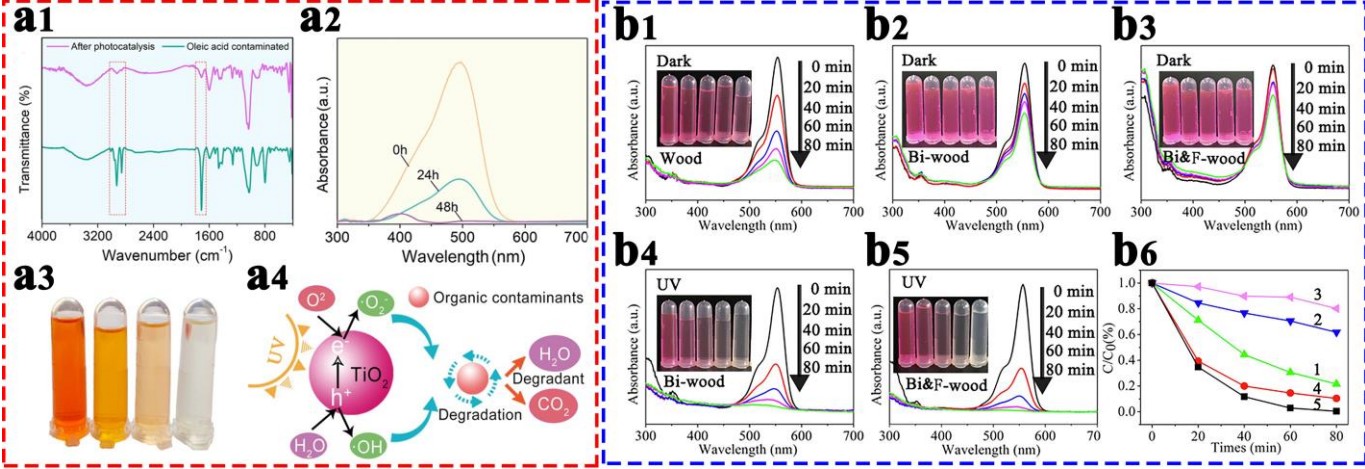

**Figure 14.** Photocatalytic functionality and efficiency of TRS-wood. (**a1**) The FTIR spectra, (**a2**) the absorbance of the methyl red solution with different irradiation times, (**a3**) the variations of methyl red solution during UV irradiation and (**a4**) the mechanism for the photodegradation of organic contaminants. Photodecomposition of RhB with (**b1**) the untreated wood, (**b2**) Bi-wood, (**b3**) Bi&F-wood in the dark, (**b4**) Bi-wood and (**b5**) Bi&F wood under UV irradiation. (**b6**) Comparison of the degradation rate of RhB with different samples as photo catalysts in the dark or under UV irradiation. Panel (**a1**–**a4**) reproduced with permission from ref. [208], copyright 2022, the authors. Panel (**b1**–**b6**) reproduced with permission from ref. [139], copyright 2019, Elsevier.

Of course, compared with many nanomaterials (i.e., Pd and Pt), the photocatalytic property of SHWSs is not good enough. This is because the photocatalytic property of SHWSs is not the main property of wood materials, which is an additional functionality that enhances the self-cleaning property. For example, wood-material-based house roofs can be functionalized with the photocatalytic property such that birds' droppings can be gradually degraded under solar irradiation.

### 3.7. Anti-Icing

Anti-icing surfaces can be defined as follows: (1) repelling cooled water droplets, (2) suppressing ice nucleation, and (3) lowering ice adhesion strength [209–211], which are suitable for SHSs as well as SHWSs. Wood materials have some differences with other materials (i.e., ceramics, metals and polymers) when discussing the anti-icing property due to their water adsorption capability. As for SHWSs, the anti-icing property can be maintained as long as the surface superhydrophobicity exists.

In cold outdoor environments, the anti-icing property of wood surfaces is very important for the utilization of wood construction [212]. This is because the mechanical property of wood materials may decrease when water-adsorbing wood materials experience freeze–thaw cycles. Thus, SHWSs are necessary to be used in wood construction to prevent wood materials from being absorbed by moisture (or water vapor). For example, Cao et al. found that the $SiO_2$/PMHOS-modified wood samples have good anti-icing properties (Figure 15a) [177]. Zhao et al. found that the hydrophobic DH-wood sample only showed a $13\% \pm 2\%$ thickness expansion (Figure 15(b1,b2)) [134]. Traditionally, water-proof paints are used for wood construction, which are good enough for regular construction. In highly

humid environments, SHWSs are recommended to be introduce onto wood constructions such that the life span of wood materials can be extended, and thus the safety of wood constructions be improved in cold and humid environments.

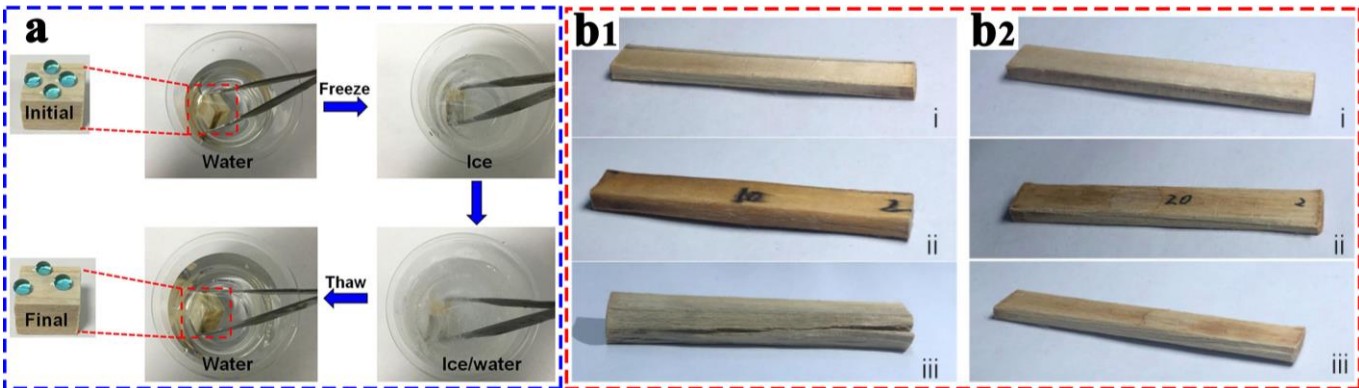

**Figure 15.** (**a**) Anti-icing test of the $SiO_2$/PMHOS-treated wood. (**b1**) Photograph of (**i**) D-wood after (**ii**) 10 and (**iii**) 30 freeze–thaw cycles. (**b2**) Photograph of (**i**) DH-wood after (**ii**) 20 and (**iii**) 50 freeze–thaw cycles. Panel (**a**) reproduced with permission from ref. [177], copyright 2022, the authors. Panel (**b1**,**b2**) reproduced with permission from ref. [134], copyright 2019, Elsevier.

### 3.8. Other Applications

Except for the above applications, SHWSs can also be utilized in many other fields, such as thermal energy storage [67,147], light-driven devices [151], photoluminescence [1,137], piezoresistive pressure sensors [49], moisture harvesting [61] and wood preservation [115], because of their excellent physiochemical properties.

When SHWSs are functionalized with light-responsive (or light-absorbing) layers, they can be utilized in photo-responsive fields. Kong et al. fabricated a polydivinylbenzene (PDVB)-nanotubes-based superhydrophobic thermal-energy-storage coating on wood substrate, and found that this coating shows a great loading capacity for IPW (78.29 wt.%) (Figure 16(a1–a3)) [67]. Yang et al. prepared a superhydrophobic TD/DW composite, and found that the calculated energy storage efficiency of the superhydrophobic TD/DW composite can reach 449.84% (Figure 16(b1–b3)) [147]. Chen et al. prepared superhydrophobic $CNT/Fe_3O_4$/MTMS wood and investigated light-driven linear motion by using NIR light (808 nm) and sunlight (Figure 16(c1,c2)) [151]. In addition to this, SHWSs can be endowed with the property of photoluminescence. For example, EI-Naggar et al. prepared a superhydrophobic nanocomposite coating on wood substrates by introducing lanthanide-doped aluminum strontium oxide nanoparticle-immobilized polystyrene [137]. Sun et al. prepared superhydrophobic wood by using $WO_3$ nanostructures via a two-step hydrothermal process [1]. Therefore, superhydrophobic coatings on wood substrates can enable wood materials as good candidates for the utilization of low-cost photo-responsive materials.

Furthermore, SHWSs have also been developed for functional devices and protective applications. For example, Li et al. studied sandwich-structured photothermal wood for durable moisture harvesting and pumping (Figure 16(d1–d3)) [61]. Huang et al. investigated superhydrophobic and high-performance wood-based piezoresistive pressure sensors for detecting human motions [49]. David et al. studied superhydrophobic coatings based on cellulose acetate for pinewood preservation [115]. Although wood materials have been applied for the above utilizations, their performances are not as good as other specially designed substrates (i.e., ceramics, metals and polymers). In a word, low-cost wood materials become promising in various fields as long as they are functionalized with surface superhydrophobicity.

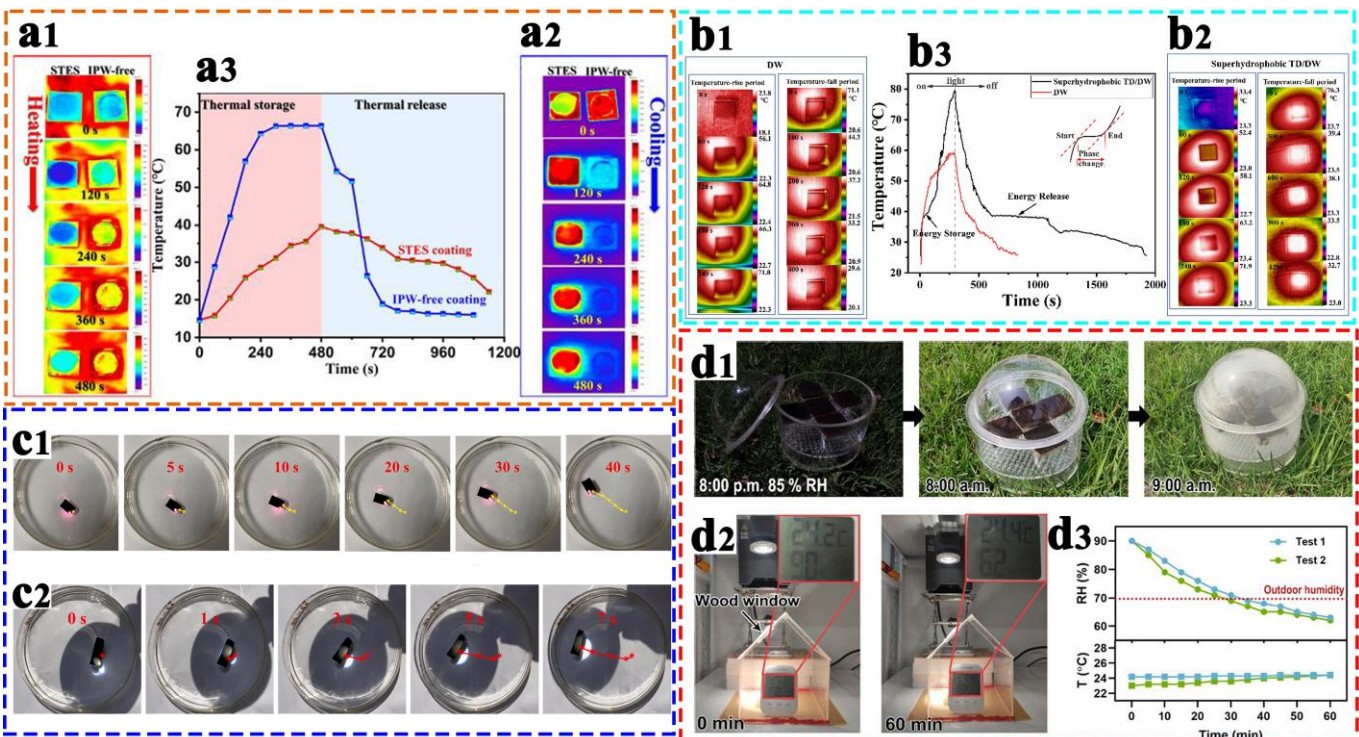

**Figure 16.** Infrared thermography images of the STES coating and IPW-free superhydrophobic coating during (**a1**) heating and (**a2**) cooling processes and (**a3**) plots of surface temperature–time. Solar-to-thermal energy conversation: infrared thermal images of DW and superhydrophobic TD/DW composites at (**b1**) temperature-rise period and (**b2**) temperature-fall period and (**b3**) temperature–time curves of DW and superhydrophobic TD/DW composites. (**c1**,**c2**) Light-driven SHWSs. Demonstrative experiments of (**d1**) moisture harvesting by a homemade apparatus and (**d2**) a house model equipped with sandwich-structured photothermal wood. (**d3**) Humidity and temperature changes in the house model. Panel (**a1**–**a3**) reproduced with permission from ref. [67], copyright 2021, American Chemical Society. Panel (**b1**–**b3**) reproduced with permission from ref. [147], copyright 2019, Elsevier. Panel (**c1**,**c2**) reproduced with permission from ref. [151], copyright 2015, The Royal Society of Chemistry. Panel (**d1**–**d3**) reproduced with permission from ref. [61], copyright 2021, American Chemical Society.

## 4. Conclusions

In this paper, we summarize recent progress in the preparation methods and the versatile practical applications of SHWSs. The preparation of SHWSs can be achieved by using various methods, including immersion, spray-coating, hydrothermal synthesis, dip-coating, deposition, sol-gel process and other methods. Each method has its own characteristics and reaction conditions. Similarly to the preparation of SHSs, the preparation of SHWSs also pursues facile, low-cost, controllable, scalable, efficient and environmentally friendly approaches. Based on the properties of wood substrates, the following issues may be considered during the preparation and practical applications of SHWSs.

(1) Multiple above preparation strategies are recommended to be combined to prepare SHWSs. The selection of preparation methods is mainly based on the experimental conditions and real practical applications. Actually, each preparation strategy usually has its own advantages and disadvantages, and sometimes an individual method is not enough to achieve SHSs on wood substrates.

(2) The decoration of nanomaterials on wood substrates for preparing SHWSs needs to consider real practical applications. The SHWSs can be endowed with many properties, such as anti-aging, thermal stability, anti-fungi/anti-bacteria, fire resistance and electromagnetic interference (EMI) shielding. However, when used in different

environmental conditions, one particular property as well as surface superhydrophobicity is mainly concerned. Thus, the selection of suitable functional nanomaterials for the preparation of SHWSs is crucial to meet the needs of special indoor and outdoor applications.

(3) The longevity of SHWSs usually determines real practical applications in the long term. The surface chemistry sustainability and the mechanical durability of SHWSs are two critical factors, which are also the challenges of SHSs. For example, an adhesive layer can be introduced to enhance the mechanical durability of SHWSs.

## 5. Challenges and Perspectives

As the global climate changes, people worry more about environmental problems and sustainable developments. For example, the term 'carbon neutralization' has drawn much attention from many fields all over the world, including research, industry and even daily life. Thus, the investigation of wood material becomes a hot issue, as wood material is a sustainable and abundant biomaterial in nature. The challenges of SHWSs are usually the challenges of SHSs, including the durability, the sustainability of low-surface-energy layers, anti-corrosion, anti-icing, anti-UV, transparency, etc. Furthermore, SHWSs also have other challenges because of the physiochemical properties of wood materials. For instance, wood materials are susceptible to fire, and thus SHWSs cannot suffer from being combusted under a high temperature for a long time. In addition, wood materials are the food of some insects and living microorganisms, which determines that wood materials may be destroyed as long as the surface superhydrophobicity of SHWSs is lost.

In the future, sustainable and abundant wood materials are promising resources for the versatile practical applications though some challenges of the utilization of wood materials still exist. When used in the fields of anti-fungi and anti-bacteria, suitable methods for preparing SHWSs should be chosen such that the surfaces (i.e., edge defects) of wood materials are all functionalized by introducing a superhydrophobic coating with proper thickness. As for oil/water separation, the pre-treatment (i.e., delignified, the preparation of wood aerogel) is necessary, which mainly increases the porous structures. After that, the resulting superhydrophobic wood substrates can possess better oil/water separation performance. Of course, the photothermal effect can also be introduced by depositing light-absorbing nanoparticles during the preparation of SHWSs. This is because the viscosity of oil/water mixtures (or emulsions) can be reduced and thus improve the separation efficiency. As for fire resistance, SHWSs cannot completely avoid fire combustion, but they can extend the time for wood materials under fire. One possible solution to enhance the fire resistance of wood materials is that flame resistant nanomaterials (metal or inorganic oxides) can be impregnated both in the inner structures as well as on the surfaces of wood materials. When used in the anti-UV field, the design of SHWSs need to consider the role of lignin that can absorb solar light. Probably, the delignified wood substrates have the promising anti-UV property after functionalizing with surface superhydrophobicity. Similarly, when wood materials are used in many other fields (i.e., EMI shielding, photocatalytic performance, anti-icing, thermal energy storage, moisture harvesting and photo-responsive devices), the SHWSs can be rationally designed based on the characteristics of practical applications, and the preparation process needs to be selected to simultaneously avoid possible drawbacks of wood materials. In summary, SHWSs functionalized from wood materials have great potential in versatile practical applications, and how to make full use of wood materials will become a hot topic in our daily life and industry.

**Author Contributions:** Conceptualization, Z.H.; investigation, X.G., M.W. and Z.H.; writing—original draft preparation, X.G., M.W. and Z.H.; writing—review and editing, Z.H.; supervision, Z.H.; project administration, Z.H.; funding acquisition, Z.H. All authors have read and agreed to the published version of the manuscript.

**Funding:** This research was funded by the Fundamental Research Funds for the Provincial Universities of Zhejiang (GK220701218030), the National Natural Science Foundation of China (Grant No. 51803043) and the Science Foundation of Hangzhou Dianzi University (KYS205620119).

**Acknowledgments:** The Fundamental Research Funds for the Provincial Universities of Zhejiang (GK220701218030), the National Natural Science Foundation of China (Grant No. 51803043) and the Science Foundation of Hangzhou Dianzi University (KYS205620119) are acknowledged for the financial support.

**Conflicts of Interest:** The authors declare no conflict of interest.

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
