# Peer review of "Superhydrophobic Wood Surfaces: Recent Developments and Future Perspectives"

_coatings, doi:10.3390/coatings13050877_

Round 1

Reviewer 1 Report

Manuscript ID: coatings-2366124 - Review

The manuscript entilted: "Superhydrophobic wood surfaces: Recent developments and future perspectives", it's a review of the recent developments and future perspectives of superhydrophobic wood surfaces (SHWSs).
I suggested some minor changes highlighted in the attached report before the publication:

1. In my opinion, the figures should be larger because you can't see the small details in the pictures.
2. Please improve the quality of the figures.
3. You can also find examples of a fire retardant composite adhesive for wood in the following citation: Zuzanna Góral, Joanna Mastalska-Popławska, Piotr Izak, Paweł Rutkowski, Joanna Gnyla, Tomasz M Majka, Krzysztof Pielichowski Impact of melamine and its derivatives on the properties of poly (vinyl acetate)-based composite wood adhesive European Journal of Wood and Wood Products 2021/1 79 177-188. Please add this citation to point 3.3.

After these minor changes, the manuscript can be accepted for publication.

English could also be polished.

Author Response

Thank you very much for reviewers’ great efforts and precious time.

Reviewer 1

The manuscript entilted: "Superhydrophobic wood surfaces: Recent developments and future perspectives", it's a review of the recent developments and future perspectives of superhydrophobic wood surfaces (SHWSs).
I suggested some minor changes highlighted in the attached report before the publication:

1. In my opinion, the figures should be larger because you can't see the small details in the pictures.

Thank you for your suggestion. We have increased the sizes of figures.

  1. Please improve the quality of the figures.

We have improved the quality of the figures (i.e., dpi 600) in the text.

  1. You can also find examples of a fire retardant composite adhesive for wood in the following citation: Zuzanna Góral, Joanna Mastalska-Popławska, Piotr Izak, Paweł Rutkowski, Joanna Gnyla, Tomasz M Majka, Krzysztof Pielichowski Impact of melamine and its derivatives on the properties of poly (vinyl acetate)-based composite wood adhesive European Journal of Wood and Wood Products 2021/1 79 177-188. Please add this citation to point 3.3.
    We have cited this article in the Section 3.3.

    After these minor changes, the manuscript can be accepted for publication.

Thank you for this positive comment.

English could also be polished.

The English language of this manuscript has been polished.

Reviewer 2 Report

The manuscript titled "Superhydrophobic wood surfaces: Recent developments and future perspectives" is well written. The following points have to be addressed before further processing of the manuscript.

1. Abstract can be included with more statements emphasizing the need for the SHWs rather than pristine woods.

2. Introduction section can be enhanced with some of the following recently published articles to describe the novelty of the review better: https://www.mdpi.com/2073-4360/14/3/589, https://pubs.rsc.org/en/content/articlehtml/2023/ta/d2ta09828h, https://www.mdpi.com/2073-4360/15/7/1682

3. A table summarizing the preparation methods of SHWs can be included after section 2.

4. A conclusion section can be included by summarizing all the discussions on the chosen topic and a few points on the author's own perspectives about the SHWs.  

Author Response

Thank you very much for reviewers’ great efforts and precious time.

Reviewer 2:

The manuscript titled "Superhydrophobic wood surfaces: Recent developments and future perspectives" is well written. The following points have to be addressed before further processing of the manuscript.

Thank you for your positive comment.

  1. Abstract can be included with more statements emphasizing the need for the SHWs rather than pristine woods.

We have revised the Abstract by adding more statements on the need for the SHWSs.

  1. Introduction section can be enhanced with some of the following recently published articles to describe the novelty of the review better: https://www.mdpi.com/2073-4360/14/3/589, https://pubs.rsc.org/en/content/articlehtml/2023/ta/d2ta09828h, https://www.mdpi.com/2073-4360/15/7/1682

These recent published articles have been cited in the Introduction Section, and the novelty of this review paper has been improved.

  1. A table summarizing the preparation methods of SHWs can be included after section 2.

A table of preparation methods for SHWSs has been added.

  1. A conclusion section can be included by summarizing all the discussions on the chosen topic and a few points on the author's own perspectives about the SHWs.

Thank you. A conclusion section has been added before the Section of Challenges and perspectives.

Reviewer 3 Report

The submitted paper provides a review on the preparation and possible applications of superhydrophobic wood surfaces (SWSs). The authors covered different methods of preparation of SWSs and highlight possible uses owing to the various properties that can be introduced by these modification processes to the wood surfaces. The paper is well-written.

Though, I have one comment. A key to the introduction of properties like anti-fungi/anti-bacteria, oil/water separation, fire-resistance, etc. to superhydrophobic wood surfaces (SWSs) is by having high contact angle (>150o) and contact angle hysteresis (<10o). I would recommend for the authors to summarize in a dedicated table the type of modification process employed (as enumerated by the authors in the early part of the manuscript), modifying material and key experimental conditions observed, type of wood surfaces used, contact angle observed, properties measured and of course, corresponding paper references.

I recommend minor revision before acceptance (see above comment).

Author Response

Thank you very much for reviewers’ great efforts and precious time.

Reviewer 3:

The submitted paper provides a review on the preparation and possible applications of superhydrophobic wood surfaces (SWSs). The authors covered different methods of preparation of SWSs and highlight possible uses owing to the various properties that can be introduced by these modification processes to the wood surfaces. The paper is well-written.

Thank you for your positive comment.

Though, I have one comment. A key to the introduction of properties like anti-fungi/anti-bacteria, oil/water separation, fire-resistance, etc. to superhydrophobic wood surfaces (SWSs) is by having high contact angle (>150o) and contact angle hysteresis (<10o). I would recommend for the authors to summarize in a dedicated table the type of modification process employed (as enumerated by the authors in the early part of the manuscript), modifying material and key experimental conditions observed, type of wood surfaces used, contact angle observed, properties measured and of course, corresponding paper references.

Thank you. We have added this type of table in the Section 2.

I recommend minor revision before acceptance (see above comment).
